# Analysis of clinically relevant variants from ancestrally diverse Asian genomes

Asian populations are under-represented in human genomics research. Here, we characterize clinically significant genetic variation in 9051 genomes representing East Asian, South Asian, and severely under-represented Austronesian-speaking Southeast Asian ancestries. We observe disparate genetic risk burden attributable to ancestry-specific recurrent variants and identify individuals with variants specific to ancestries discordant to their self-reported ethnicity, mostly due to cryptic admixture. About 27% of severe recessive disorder genes with appreciable carrier frequencies in Asians are missed by carrier screening panels, and we estimate 0.5% Asian couples at-risk of having an affected child. Prevalence of medically-actionable variant carriers is 3.4% and a further 1.6% harbour variants with potential for pathogenic classification upon additional clinical/experimental evidence. We profile 23 pharmacogenes with high-confidence gene-drug associations and find 22.4% of Asians at-risk of Centers for Disease Control and Prevention Tier 1 genetic conditions concurrently harbour pharmacogenetic variants with actionable phenotypes, highlighting the benefits of pre-emptive pharmacogenomics. Our findings illuminate the diversity in genetic disease epidemiology and opportunities for precision medicine for a large, diverse Asian population.

Genomics is increasingly an integral part of mainstream medicine and has the potential to revolutionize healthcare delivery globally[1]. A critical enabler of precision medicine is the availability of genomic variation data from both patients and the general population, to accurately assess whether a variant is disease-causing and to identify genetic disorders prevalent in the population[2]. Despite advances in genomics research, persistent Eurocentric biases in sequencing studies have resulted in inequitable access to precision medicine[3–5]. Although comprising nearly 60% of the global population, Asian genomes are relatively scarce; for example, constituting 6.6% of the widely-used Genome Aggregation Database (gnomAD, v3.1) and 3% of population health studies[6,7]. Despite the diversity among Asians[8], nearly all Asian genomes in population databases are of East and South Asian ancestry, with severe under-representation of Southeast Asians.

Increasing Asian representation in the characterization of medically relevant population genetic data is crucial to address several disparities that affect a large global population. First, healthcare professionals serving non-European populations may be less aware of

genetic disorders and associated symptoms in their patients, increasing risk of misdiagnosis or mistreatment[9]. Second, carrier screening panels are mostly derived from European-descent populations and may miss genetic disorders common in non-Europeans. Finally, bias in submissions to variant databases leads to the clinical interpretation of rare variants in non-Europeans being more challenging, reducing the likelihood of reporting and perpetuating the lack of publicly available information[10,11]. Emerging work on diverse populations is also highlighting complex relationships between self-reported and genetically-inferred ancestry, reflecting the importance of considering admixture when evaluating personalized genetic risk[12]. This is relevant given the spread and integration of the Asian diaspora with other continental populations.

Singapore, a Southeast Asian city-state of four million residents[13], has a diverse population comprising three major ethnic groups: Chinese (74.2%), Malay (13.7%), Indian (8.9%) of East Asian, Southeast Asian, and South Asian ancestry, respectively. Population-scale sequencing of Singaporean genomes is thus a particularly attractive

✉e-mail: patrick.tan@precise.cris.sg; Saumya.S.Jamuar@singhealth.com.sg; joanne.ngeow@ntu.edu.sg; wengkhong.lim@duke-nus.edu.sg

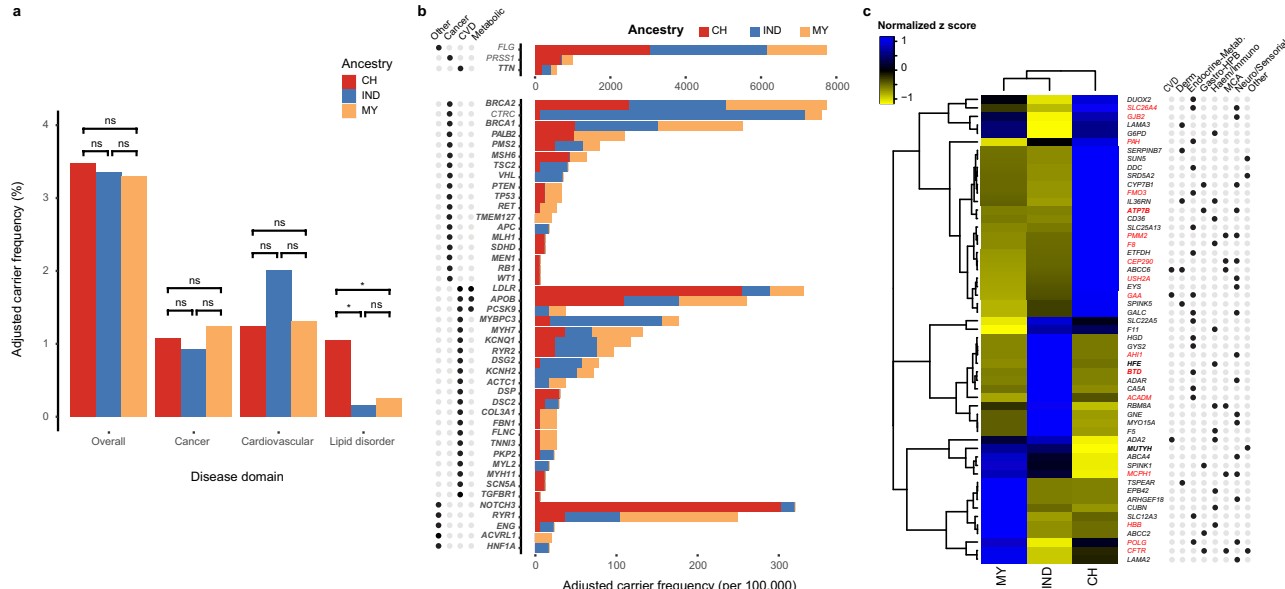

**Fig. 1 | Spectrum of pathogenic variation in clinically relevant genes among Singaporeans. a** Carrier frequencies of ACMG SF v3.0 genes associated with dominant disorders compared across the three main ancestry groups. The disorders are further sub-classified into three main disease domains for comparison (cancer, cardiovascular, lipid disorders). Carrier frequency of P/LP variants in lipid disorder genes were significantly higher among Chinese compared to Indians ($p = 7.93 \times 10^{-5}$) and Malays ($p = 1.70 \times 10^{-3}$). Statistical significance was evaluated by two-sided Fisher's exact test, with Benjamini-Hochberg correction for multiple testing. Adjusted $p < 0.05$ was considered significant, ns: not significant. CH: Chinese, IND: Indian, MY: Malay. **b** Differential distribution of carrier frequencies across ancestries for dominantly inherited genetic disorders associated with ACMG SF v3.0 medically actionable genes, or non-ACMG SF v3.0 genes with a carrier frequency >0.5%. **c** Genes of recessive conditions with significant differences in

carrier frequency of P/LP variants across ancestry groups. Colour scale maps to row-wise z-scores, obtained by subtracting from each gene-level carrier frequency the row average and then dividing the value by the row standard deviation. Genes in red fonts are recommended by ACMG for carrier screening. Genes in bold fonts are part of the ACMG SF v3.0 list. The disorder domain associated with pathogenic alteration of the indicated gene is represented in the dot matrix. CVD cardiovascular disorders, Derm dermatological disorders, Metab. metabolic (including lysosomal storage, mitochondrial, metabolic disorders), Gastro-HPB gastro-hepato-pancreato biliary disorders, Haem/Immuno haematological/immunological disorders, MCA multiple congenital anomalies, Neuro neurological (including neurologic, neuromuscular, neurodegenerative disorders), Others: including cancer, respiratory, genitourinary disorders.

effort[14] to provide insights into genetic disease risk and to address knowledge gaps for populations across East Asia, South Asia, and a major proportion of Austronesian-speaking Southeast Asian group represented by Malays.

Here, we perform deep interrogation of clinically significant genetic variants from 9051 Singaporean whole genomes and characterize (1) prevalence of autosomal dominant (AD) disorders, (2) carrier frequency of autosomal recessive (AR) and X-linked conditions, and (3) evaluate distribution of pharmacogenomic variation across the three ancestry groups. We also examine the implications of genetic admixture on personalized disease risk in this ancestry-diverse population. Our findings demonstrate the diversity of genetic epidemiology of disease in multi-ethnic Asian populations and highlight opportunities for coupling genetic disease risk profiling with pre-emptive pharmacogenomics for therapy optimization.

## Results

### Study characteristics

Our analysed cohort of 9,051 individuals from SG10K_Health project is a cross-section of the Singaporean population, inferred to be unrelated to the second degree (Supplementary Table 1). Individual age ranged between birth-85 years (median: 47 years) and comprised 57.3% females (Supplementary Table 2). Using ADMIXTURE (ver 1.3.0)[15], we inferred genetic ancestry of individuals, who were mostly Chinese (60.8%) followed by Indian (21.4%) and Malay (17.8%). Whole genome sequences were jointly analysed and variants occurring in 4,143 genes associated with AD, AR and X-linked monogenic disorders were curated according to the American College of Medical Genetics and Genomics (ACMG) guidelines and classified according to a standardized workflow (Supplementary Fig. 1). Overall, we identified 4,960

pathogenic/likely pathogenic (P/LP) single nucleotide variants and micro-indels, of which 82.2% were protein-length changes, as well as 406 gross deletions in loss-of-function intolerant (LOFi) genes.

### Prevalence of variants associated with autosomal dominant disorders

We identified 238 (2.63%) individuals harbouring at least one of the 163 P/LP variants identified in 35 dominant condition genes of the ACMG secondary findings (SF) v2.0 gene list (Supplementary Data 1, 2), the prevalence of which is comparable to reported yields of 1.86% to 2.85% in smaller East Asian cohorts[16,17] and 2.0% to 2.54% predominantly European cohorts[18,19]. This yield increased to 3.41% with the expanded ACMG SF v3.0 list, identifying an additional 71 individuals, most of whom are protein-truncating variant carriers in the newly included cardiomyopathy gene *TTN* (53/71) and hereditary breast and ovarian cancer (HBOC) syndrome gene *PALB2* (11/71). Only two individuals (2/309) had P/LP variants in multiple genes: one harbouring predisposition to familial hypercholesterolemia (FH) and long QT syndromes (*PCSK9, KCNH2*) and the other with predisposition to cancer and hypertrophic cardiomyopathy (*SDHD, MYBPC3*).

Although the overall prevalence of AD disorder variants across ancestry groups was similar ($p > 0.05$), concentration of genetic risk was unequal for certain disease domains (Fig. 1a, Supplementary Table 3). Notably, we observed significantly higher genetic risk for FH among Chinese (1.05%) compared to Indians (0.15%, $p = 7.93 \times 10^{-5}$) and Malays (0.25%, $p = 1.70 \times 10^{-3}$), predominantly driven by *LDLR* carriers among Chinese (0.76%, Table 1). While genetic risk for cancer and cardiovascular disorders were not significantly different across the three ancestry groups, we found ancestry-specific distinctions at the variant level. For instance, carrier frequency for P/LP variants in the

**Table 1 | Consolidated top 10 autosomal dominant and autosomal/X-linked recessive disorder genes with highest carrier frequencies of P/LP variants identified in each ancestry group in Singapore**

| Gene | Associated conditions | Adjusted carrier frequency | | | | | |
|---|---|---|---|---|---|---|---|
| | | Chinese (*n* = 5502) | | Indian (*n* = 1941) | | Malay (*n* = 1608) | |
| | | No. carriers | (%) | No. carriers | (%) | No. carriers | (%) |
| Dominant disorder genes | | | | | | | |
| *FLG* | Ichthyosis vulgaris | 505 | (9.18) | 181 | (9.33) | 76 | (4.73) |
| *PRSS1* | Hereditary pancreatitis | 116 | (2.11) | 1 | (0.05) | 14 | (0.87) |
| *NOTCH3* | CADASIL (Cerebral Autosomal Dominant Arteriopathy with Sub-cortical Infarcts and Leukoencephalopathy), Infantile myofibromatosis | 50 | (0.91) | 1 | (0.05) | 0 | (0) |
| *LDLR* | Familial hypercholesterolemia | 42 | (0.76) | 2 | (0.10) | 2 | (0.12) |
| *TTN* | Dilated cardiomyopathy | 33 | (0.60) | 13 | (0.67) | 7 | (0.44) |
| *BRCA2* | Hereditary breast and ovarian cancer | 19 | (0.35) | 7 | (0.36) | 6 | (0.37) |
| *APOB* | Familial hypercholesterolemia | 18 | (0.33) | 4 | (0.21) | 4 | (0.25) |
| *BRCA1* | Hereditary breast and ovarian cancer | 8 | (0.15) | 6 | (0.31) | 5 | (0.31) |
| *PALB2* | Hereditary breast and ovarian cancer | 8 | (0.15) | 0 | (0) | 3 | (0.19) |
| *MSH6* | Lynch syndrome | 7 | (0.13) | 0 | (0) | 1 | (0.06) |
| *RYR1* | Malignant hyperthermia susceptibility | 6 | (0.11) | 4 | (0.21) | 7 | (0.44) |
| *MYH7* | Hypertrophic cardiomyopathy, Dilated cardiomyopathy | 6 | (0.11) | 2 | (0.10) | 3 | (0.19) |
| *KCNQ1* | Long-QT syndrome type 1 | 4 | (0.07) | 3 | (0.15) | 2 | (0.12) |
| *RYR2* | Catecholaminergic polymorphic ventricular tachycardia | 4 | (0.07) | 3 | (0.15) | 1 | (0.06) |
| *MYBPC3* | Hypertrophic cardiomyopathy | 3 | (0.05) | 8 | (0.41) | 1 | (0.06) |
| *CTRC* | Hereditary pancreatitis | 1 | (0.02) | 19 | (0.98) | 1 | (0.06) |
| *DSG2* | Arrhythmogenic right ventricular cardiomyopathy | 1 | (0.02) | 3 | (0.15) | 1 | (0.06) |
| *KCNH2* | Long-QT syndrome type 2 | 0 | (0) | 3 | (0.15) | 1 | (0.06) |
| Recessive disorder genes | | | | | | | |
| *GJB2* | Autosomal recessive deafness | 1094 | (19.88) | 70 | (3.61) | 256 | (15.92) |
| *CFTR* | Cystic fibrosis, congenital bilateral absence of vas deferens, hereditary pancreatitis | 458 | (8.32) | 68 | (3.5) | 264 | (16.42) |
| *CD36* | Platelet glycoprotein IV deficiency | 328 | (5.96) | 1 | (0.05) | 4 | (0.25) |
| *HFE* | Haemochromatosis | 310 | (5.63) | 325 | (16.74) | 79 | (4.91) |
| *IL36RN* | Pustular psoriasis | 191 | (3.47) | 2 | (0.10) | 10 | (0.62) |
| *DUOX2* | Thyroid dyshormonogenesis | 178 | (3.24) | 26 | (1.34) | 38 | (2.36) |
| *G6PD* [a] | Glucose-6-phosphate dehydrogenase deficiency | 162 | (2.94) | 32 | (1.65) | 46 | (2.86) |
| *SLC25A13* | Citrullinemia type II | 126 | (2.29) | 2 | (0.10) | 0 | (0) |
| *POLG* | Mitochondrial DNA depletion syndrome (Alpers type, MNGIE type), mitochondrial recessive ataxia syndrome (MIRAS), progressive external opthalmoplegia (PEO) | 116 | (2.11) | 5 | (0.26) | 53 | (3.30) |
| *SLC26A4* | Autosomal recessive deafness, Pendred syndrome | 109 | (1.98) | 13 | (0.67) | 17 | (1.06) |
| *SERPINB7* | Palmoplantar keratoderma (Nagashima type) | 109 | (1.98) | 1 | (0.05) | 3 | (0.19) |
| *HBB* | Beta-thalassemia | 94 | (1.71) | 23 | (1.18) | 123 | (7.65) |
| *SPINK1* | Hereditary pancreatitis | 83 | (1.51) | 85 | (4.38) | 100 | (6.22) |
| *SLC22A5* | Systemic primary carnitine deficiency | 77 | (1.40) | 43 | (2.22) | 5 | (0.31) |
| *ABCA4* | Stargardt disease, Retinitis pigmentosa | 52 | (0.95) | 60 | (3.09) | 69 | (4.29) |
| *MYO15A* | Autosomal recessive deafness | 40 | (0.73) | 55 | (2.83) | 17 | (1.06) |
| *GNE* | Nonaka myopathy | 19 | (0.35) | 66 | (3.40) | 13 | (0.81) |
| *ARHGEF18* | Retinitis pigmentosa | 16 | (0.29) | 6 | (0.31) | 31 | (1.93) |
| *BTD* | Biotinidase deficiency | 15 | (0.27) | 144 | (7.42) | 7 | (0.44) |
| *F5* | Factor V deficiency | 7 | (0.13) | 49 | (2.52) | 8 | (0.50) |

[a] X-linked recessive gene.
Carrier frequencies were adjusted to the total individuals in each ancestry group.

hypertrophic cardiomyopathy gene *MYBPC3* was eight-fold higher among Indians (0.41%) compared to Chinese (0.05%), attributed to the significantly higher frequency of *MYBPC3* c.1790G > A (p.Arg597Gln) variant (Indian: 0.31% vs Chinese: 0%, $p = 9.38 \times 10^{-4}$). We also observed significantly higher carrier frequency of a known Malay founder variant associated with HBOC[20], *BRCA1* c.2726dup (p.Asn909Lysfs*6) among Malays in our study (0.25%, $p = 0.032$) compared to Chinese (0.02%) (Supplementary Fig. 2). To account for potential survivorship bias in our observation, we quantified carrier frequencies for our cohort subset aged ≤ 50 years and found that these ancestry-specific distinctions remain significant (Supplementary Data 3).

Beyond ACMG SF v3.0 genes, we identified four AD genes (*FLG*, *NOTCH3*, *PRSS1*, *CTRC*) with carrier frequencies exceeding 0.5% in at least one ancestry group (Table 1). These genes are either associated with non-life-threatening disorders (*FLG*; ichthyosis vulgaris), late-onset disorders (*NOTCH3*; cerebral autosomal dominant arteriopathy with sub-cortical infarcts and leukoencephalopathy, CADASIL) or risk factors for disease (*PRSS1*, *CTRC*; hereditary pancreatitis). Genetic risk

differed across ancestry groups for these genes (Fig. 1b), primarily driven by ancestry-specific recurrent variants. For instance, CADASIL risk among Chinese stems from a recurrent *NOTCH3* c.1630C > T (p.Arg544Cys) variant (0.91%) also prevalent among Taiwanese[21], whereas the underlying genetic risk for hereditary pancreatitis differed between Chinese and Indians, contributed by a Chinese-predominant *PRSS1* c.623G > C (p.Gly208Ala) variant (1.94%) and Indian-specific *CTRC* c.217G > A (p.Ala73Thr) variant (0.98%), respectively (Supplementary Data 2). Overall, carrier frequencies for genes with burden exceeding 0.5% in Chinese or Indians correlated well with frequencies in gnomAD East Asian and South Asian populations respectively (Supplementary Fig. 3, Pearson's $r = 0.93$, $p = 3.8 \times 10^{-22}$).

## Carrier frequencies of variants associated with autosomal and X-linked recessive conditions

Next, we evaluated the population carrier burden of recessive conditions. Among AR genes, high carrier burden was observed for *GJB2*, *CFTR*, and *HFE* (Table 1), each driven by elevated carrier frequencies in specific variants that confer milder disease. For instance, we detected a predominant *GJB2* variant among Chinese and Malays; c.109G > A (p.Val37Ile; Chinese:18.5%, Malay:15.1%), known to be associated with mild-to-moderate hearing impairment[22], whereas the *HFE* c.187C > G (p.His36Asp) variant identified recurrently among Indians (16.6%) has rarely been associated with frank clinical hemochromatosis although the variant is linked to biochemical abnormalities[23]. Despite high *CFTR* carrier burden, the variants with high carrier frequencies, c.4056G > C (p.Gln1352His) and c.1210-11T > G, are associated with congenital bilateral absence of vas deferens (CBAVD) and pancreatitis instead of cystic fibrosis. Nevertheless, we observed a few genes with high carrier burden that are driven by high carrier frequencies in known causal variants for disorders, such as the significant burden of *SLC25A13* in Chinese (2.29%, $p = 2.51 \times 10^{-14}$) due to a high carrier frequency of citrin deficiency-linked variant *SLC25A13* c.852_855del (p.Met285Profs*2)[24,25] (Chinese: 1.45%) and *GNE* in Indians (3.40%, $p < 9.6 \times 10^{-8}$) attributed to the GNE myopathy-linked c.2086G>A (p.Val696Met) variant[26] (Indian: 3.40%).

Comparing disease risk profiles across ancestry groups, we observed distinctions attributable to highly recurrent variants in different genes (Fig. 1c). Among Malays, who are unrepresented in existing population databases, we found higher carrier burden for beta-thalassemia, contributed by the common Southeast Asian *HBB* c.79G > A (p.Glu27Lys; 6.72% Malays), and retinopathies driven by recurrent variants in retinopathy-related genes *ABCA4* (Stargardt disease, c.71G > A (p.Arg24His), 2.36%) and *ARHGEF18* (retinitis pigmentosa, c.826-1G > A, 1.80%). Enriched among Chinese were recurrent variants in immune-related disorders, namely platelet glycoprotein IV deficiency-associated *CD36* c.332_333del (p.Thr111Serfs*22, 3.29%) and generalized pustular psoriasis-linked *IL36RN* c.115 + 6 T > C (3.18%), as well as Krabbe leukodystrophy-associated *GALC* c.1901T > C (p.Leu634Ser, 1.15%); all of which are prevalent disease-associated variants reported in East Asian populations[27–29]. In Indians, we observed a high carrier frequency of factor V deficiency-associated *F5* 'Leiden' c.1601G > A (p.Arg534Gln) variant (2.27% Indians) and a high carrier burden in *BTD* (7.42% Indians), which is predominantly driven by c.1270G > C (p.Asp424His; 6.80% Indians), a known mild variant that causes partial biotinidase deficiency in conjunction with another severe *BTD* variant[30]. Other recessive genes with carrier frequencies exceeding 1% include those associated with Pompe disease (*GAA*), Shwachman-Diamond syndrome (*SBDS*), *EYS*-associated retinitis pigmentosa, Gitelman syndrome (*SLC12A3*), and *DUOX2*-associated congenital hypothyroidism (Supplementary Data 1).

### Gaps in carrier screening panels for Asians
Next, we evaluated the coverage of existing carrier screening panel recommendations against our population carrier burden of recessive conditions. We identified 70 recessive genes with carrier frequencies exceeding 0.5% in at least one ancestry (Supplementary Data 1), of which 21 genes are recommended by ACMG for carrier screening[31], a further 18 genes are covered by commercial carrier screening panels, and the remaining 31 genes provided scope for expansion of carrier screening panels to better represent genetic disorders in Asian populations.

Among the 70 genes, 37 are associated with severe recessive diseases, defined as "conditions with lethality in childhood, are significantly disabling or have a negative impact on quality of life for an affected child and the family"[32]. Ten of these 37 genes (27%) warranted inclusion but are not found in commercial carrier screening panels. These genes are associated with metabolic (*DDC, GYS2*), cardiovascular (*ABCC6*), developmental (*SBDS*), neurodegenerative (*ADAR*), ocular (*ABCA4*), respiratory (*DNAH11*), gastroenterologic (*CYP7B1*), immunological (*ADA2*) and dermatological (*SPINK5*) disorders. Additionally, we estimated the proportion of couples in each ancestry group potentially at risk of having offspring affected by AR disorders (at-risk couples, ARCs) by exhaustively simulating all possible matings and then identifying instances where both partners in a theoretical pairing carry a P/LP variant in the same gene. Considering only 1,300 genes that cause severe recessive disorders[32], we detected ARC proportions of 0.70%, 0.56%, and 0.51% in Malays, Indians and Chinese, respectively.

### Gross deletions in loss-of-function intolerant (LOFi) genes
To determine the contribution of gross deletions to genetic disease risk, we identified pathogenic deletions between 500 bases to 10 megabases (Mb) affecting biologically relevant transcripts of LOFi genes. We found clinically significant deletions affecting *SMN1* (AR spinal muscular atrophy) in 1.92% (37/1,923) of individuals and the 19 kb *HBA1/HBA2* SEA deletion linked to alpha-thalassemia at a carrier frequency of 1.16% (Supplementary Data 4). We also detected a 2.9 kb deletion in *AGT* (AR renal tubular agenesis) previously reported as a Taiwanese founder mutation[33] and a 3.2 kb deletion in *CLMP* (AR congenital short bowel syndrome) in 0.61% and 0.20% of Chinese individuals, respectively. Among Indians, recurrent pathogenic deletions include *CNGA1* (15 kb deletion, retinitis pigmentosa, 0.31%) and *ALMS1* (1.3 kb deletion, Alström syndrome, 0.16%), whereas pathogenic deletions found in Malays include *IFT140* (4.2 kb deletion, Mainzer-Saldino syndrome, 0.31%) and *SLURP1* (32 kb deletion, Mal de Meleda, 0.31%).

### Genetic ancestry mapping reveals limitations of self-reported race/ethnicity (R/E)
The use of self-reported R/E for evaluating genetic disease risk has implications in a multi-ethnic population such as Singapore because it is a social construct that does not reliably capture one's genetic ancestry. To assess this effect, we compared population demography defined by self-reported R/E (captured in individual national identification document) with genetic ancestry inferred using ADMIXTURE fitted to three hypothetical ancestral components (K = 3), which recapitulated the three major ancestry groups in SG10K_Health (Fig. 2a). Two groups emerged; individuals whose self-reported R/E was inconsistent ('R/E-mismatched group', $n = 268$, Supplementary Table 4) or consistent ('R/E-matched group', $n = 8783$) with the predominant ancestral component assigned by ADMIXTURE. Using the highest ancestral component proportion, maxQ, as a measure of admixture (with lower maxQ indicating higher admixture), we found that the R/E-mismatched group had significantly lower median maxQ compared to R/E-matched group (0.53 vs. 0.87, $p = 1.93 \times 10^{-89}$), implying that recent admixture (e.g., mixed parentage), may be prevalent among R/E-mismatched individuals (Supplementary Fig. 4).

Given admixture in the population, it is conceivable individuals may harbour clinically significant variants highly specific to other ancestries ('discordant carriers'). Using local ancestry inference, we

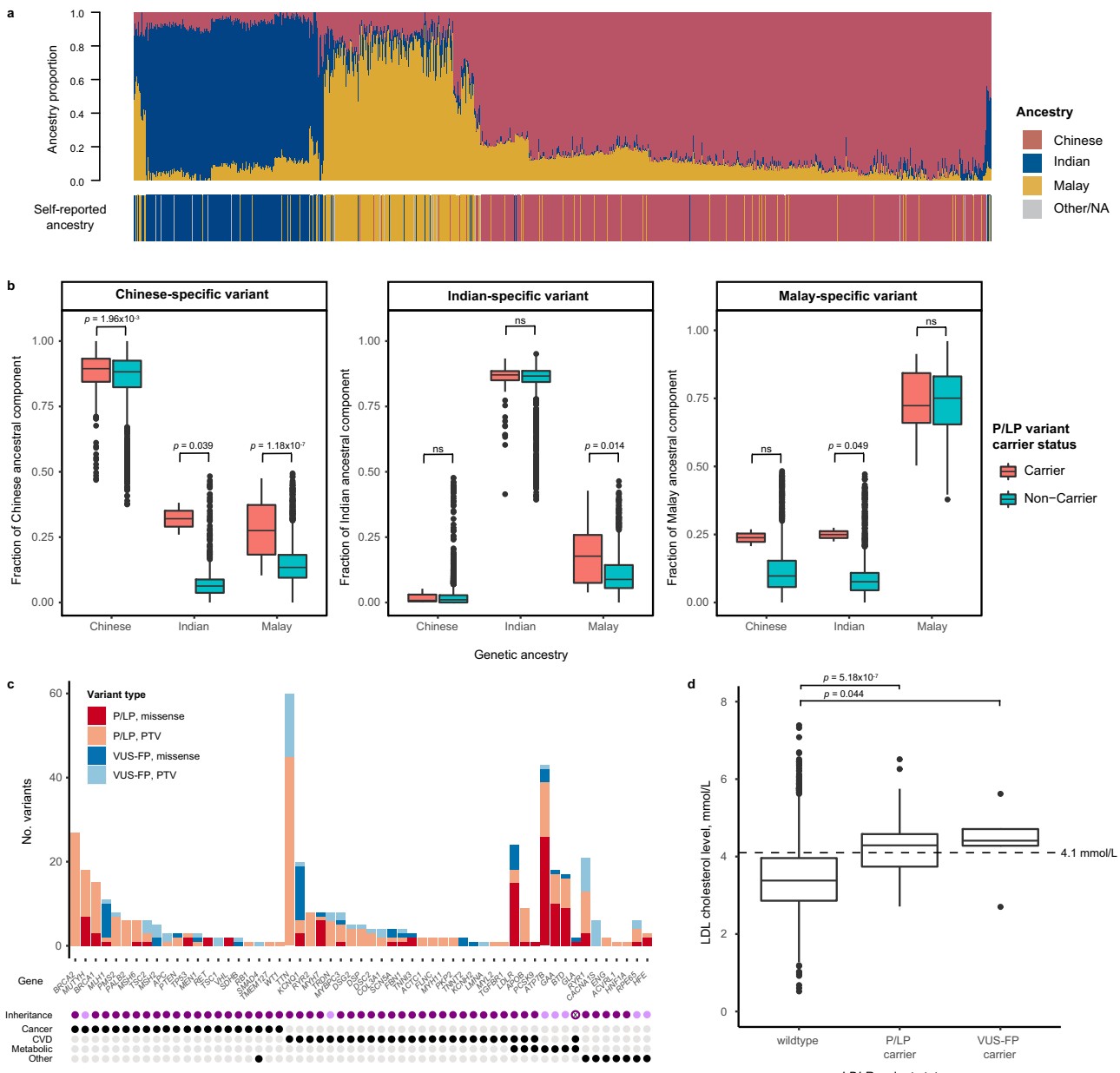

**Fig. 2 | Evaluating the influence of genetic admixture and potentially pathogenic VUS in SG10K_Health cohort. a** ADMIXTURE analysis of inferred genetic ancestral components at K = 3 juxtaposed with self-reported ancestry for the 9,051 Singaporean individuals. NA: not available. **b** The proportion of genetic ancestral components tracked consistently with carrier status of pathogenic/likely pathogenic (P/LP) variants specific to the associated ancestry group across Singaporean Chinese (CH), Indian (IND) and Malay (MY) individuals (Chinese-specific variant carriers/non-carriers: (CH) 455/5047, (IND) 2/1939, (MY) 26/1582; Indian-specific variant carriers/non-carriers: (CH) 3/5499, (IND) 147/1794, (MY) 19/1589; Malay-specific variant carriers/non-carriers: (CH) 2/5500, (IND) 2/1939, (MY) 24/1584). Pairwise differences between carriers and non-carriers were evaluated by two-sided Wilcoxon rank-sum test. *p* < 0.05 was considered significant. ns: not significant. **c** Juxtaposition of P/LP variants with potentially pathogenic variants of uncertain significance (missense and cryptic splice variants) classified as VUS-FP, identified in

genes from the ACMG SF v3.0 list. Mode of inheritance and disease domain associated for each gene are indicated in the dot matrix below. PTV: protein-truncating variant. **d** Carriers of VUS-FP variants (*n* = 5) identified in *LDLR* demonstrated LDL cholesterol range that is consistent with carriers of P/LP variants (*n* = 25) and is higher compared to non-carriers (*n* = 4397). An LDL cholesterol level of ≥4.1 mmol/L is classified as high by the Ministry of Health Singapore. *p* values were derived from binomial logistic regression comparing LDL cholesterol levels against *LDLR* variant status, correcting for age, sex, genetic ancestry, and lipid-lowering medication intake. All box plots extend from the 25th to 75th percentiles and the length of the whiskers are defined as follows: upper whisker = min(maximum_value, Q3 + 1.5*IQR), lower whisker = max(mininum_value, Q1−1.5*IQR), where IQR is interquartile range, Q3 is third quartile, Q1 is first quartile. Horizontal line in the box represents the median.

identified 177 variants that are exclusive to one ancestral population ('ancestry-specific variant'), of which 37 were found in 54 discordant carriers. The majority of discordant carriers were R/E-matched (52/54), suggesting cryptic admixture. We found discordant carriers harboured more of the ancestral component linked to the ancestry-specific variant (Fig. 2b, pink bars) than non-carriers for all three ancestral

components investigated. For example, the Chinese ancestral component was significantly higher among Indian and Malay carriers of a Chinese-specific variant compared to non-carriers (Fig. 2b left panel, Supplementary Table 5), with a median Chinese ancestral component between 28%-32% that is supportive of cryptic admixture. Overall, we were more likely to detect discordant variants (odds ratio (OR): 5.6,

95% confidence interval (CI): 3.11–10.38, $p = 6.98 \times 10^{-10}$, two-sided Fisher's exact test) among individuals with higher levels of genetic admixture (i.e. individuals in the lowest quartile of maxQ values within their ancestry group).

## Estimates of pathogenic potential among variants of uncertain significance (VUS)

Given that deleterious Asian variants are likely to be under-reported or unreported in clinical databases such as ClinVar[34], we sought to explore potentially pathogenic variants that did not meet our P/LP classification criteria among VUS. We identified missense and cryptic splicing variants with predicted deleterious outcomes using in silico criteria, which we designated as VUS-favour pathogenic (VUS-FP). Among 20,867 VUS with prediction scores, we detected 639 VUS-FP, of which 472 (73.9%) were not reported in ClinVar. Of these, 106 variants occurred in the ACMG SF v3.0 gene list (Supplementary Data 5) and we identified an additional 148 individuals with dominantly inherited conditions, translating to an estimated increase in the prevalence of AD conditions in our cohort from 3.41% to 5.05%. We showed that gene-level distribution of variant type tracked the spectrum for known pathogenic variants (Fig. 2c); for instance, missense VUS-FP were predominantly identified in *LDLR* and *KCNQ1*, genes in which missense variants account for half of the reported disease-associated variants.

Using *LDLR* variants and available low-density lipoprotein (LDL) cholesterol measurements, we evaluated the pathogenicity of VUS-FP. We found that individuals harbouring P/LP and VUS-FP variants were more likely to have clinically high LDL cholesterol levels (defined as ≥4.1 mmol/L by the Ministry of Health Singapore) compared to non-carriers (Fig. 2d), even after adjusting for age, sex, ancestry and lipid-lowering medication intake (P/LP: OR = 10.83, 95%CI = 4.52–30.05, $p = 5.18 \times 10^{-7}$; VUS-FP: OR = 9.67, 95%CI = 1.41–190.62, $p = 0.044$). This corroborated our in silico assessment of *LDLR* VUS-FP, suggesting that VUS-FP account for a proportion of "missing pathogenicity"[35] in underrepresented populations.

## Pharmacogenomic landscape and interaction with genetic disease risk

Beyond genetic disease risk, understanding pharmacogenomic diversity, that is variation in the frequency of alleles known to alter an individual's response to medication, has clinical implications. To examine the pharmacogenomic landscape, we identified known pharmacogenetic alleles of genes in the Clinical Pharmacogenetics Implementation Consortium (CPIC) drug-gene pair list with Pharmacogenomics Knowledgebase (PharmGKB) level 1 evidence. Collectively, 99.7% (9,026/9,051) of SG10K_Health individuals carried at least one actionable pharmacogenetic finding in 23 pharmacogenes with high-confidence gene-drug associations, with a median of five findings per individual. This high frequency is predominantly due to carriers (>98%) of *VKORC1* c.−1639G > A (rs9923231) allele affecting sensitivity to the anticoagulant warfarin, which is known to be prevalent among Asians[36]. Of 154 pharmacogenetic variants with actionable phenotype identified (Supplementary Data 6), 76.6% (118/154) had a minor allele frequency (MAF) < 1% and 31.8% (49/154) were very rare variants carried by only 1–2 individuals, over half (57.1%, 28/49) of which were found in genes of the cytochrome P450 CYP2 family. Over one-quarter (26.8%, 2429/9051) of our cohort carried a genotype associated with life-threatening drug toxicities including allopurinol- or carbamazepine-induced Stevens-Johnson syndrome/ toxic epidermal necrolysis (SJS/TEN, 25.6% *HLA-A* or *HLA-B* risk allele carriers), DPD deficiency-linked fluorouracil toxicity (1.4% *DPYD* intermediate or poor metabolizers) and malignant hyperthermia susceptibility due to potent volatile anaesthetic agents and succinylcholine (0.07% *CACNA1S* or *RYR1* risk allele carriers).

Overall, we observed that individuals with actionable pharmacophenotypes associated with commonly prescribed drugs were relatively prevalent, irrespective of ancestry (Table 2). Notably, high fractions of individuals were identified with a genotype affecting the activity of cytochrome P450 family of enzymes (Supplementary Data 7); for instance 51.0%-77.2% individuals across ancestries harboured alleles associated with actionable phenotypes in *CYP2C19*, which is important for metabolism of widely used drugs including the antiplatelet clopidogrel, antiemetics (proton pump inhibitors) and antidepressants such as selective serotonin uptake inhibitors (SSRIs), whereas 31.1–47.2% individuals carried actionable pharmacogenetic variants in *CYP2D6* for a broad range of drug interactions including opioids, antidepressants, and tamoxifen therapy for cancer. However, we also found that the prevalence of certain pharmacophenotypes was variable by ancestry; for instance, there were significantly more poor metabolizers among Indians (17.4%) compared to Chinese (3.2%, $p = 7.28 \times 10^{-66}$) and Malays (1.3%, $p = 6.50 \times 10^{-51}$) for *UGT1A1*, which metabolizes irinotecan-based drugs frequently used in cancer treatments, due to a higher allele frequency of *UGT1A1*28* among Indians. Ancestry-specific variability may also underlie differential genetic profiles for sensitivity to warfarin, which can be attributed to the high frequency *VKORC1* rs9923231 among Chinese and Malays as well as the *CYP4F2* rs2108622 (c.1297G > A, p.Val433Met) and *CYP2C9*3* alleles especially prevalent among Indians (Supplementary Data 6).

Next, we explored the intersection of individual genetic disease risk with pharmacogenomic profile by estimating the frequency of individuals harbouring pharmacogenetic variants associated with an actionable phenotype to drugs used for the disorder they are genetically predisposed to. We identified 143 individuals at risk of Centers for Disease Control and Prevention (CDC) Tier 1 genetic conditions (HBOC, Lynch syndrome, FH)[37], of whom 32 (22.4%) concurrently harboured a pharmacogenetic variant with actionable phenotype to drugs commonly used for treatment of their condition (Fig. 3, Supplementary Table 6). Specifically, 23.0% (14/61) of individuals susceptible to HBOC were also *CYP2D6* intermediate or poor metabolizers, who are at higher risk of therapeutic failure for tamoxifen and breast cancer recurrence, whereas eight among 17 individuals with Lynch syndrome predisposition carried either a *UGT1A1*6* or *UGT1A1*28* allele associated with toxicities related to irinotecan-based chemotherapy. Finally, 15.4% (10/65) of FH-predisposed individuals are concurrently at risk of statin drug-induced myopathies attributed to *SLCO1B1* c.521T > C (p.Val174Ala, rs4149056) variant and would benefit from dose adjustment or alternative statins[38].

To evaluate for potentially deleterious novel pharmacogenetic variants, we curated for loss-of-function (LOF) variants in 10 of our list of 23 pharmacogenes, whereby LOF is the mechanism associated with actionable phenotype. We identified 47 putative LOF variants, all with a MAF less than 1%. Over half (33/47, 70.2%) of these putative LOF variants are rare, occurring as singletons or doubletons (Supplementary Data 8), consistent with the proportions of singleton-doubleton LOF variants reported in whole genome/exome studies from other populations (>58%)[39,40]. Notably, half (25/47, 53.2%) of the putative LOF variants were found within the highly polymorphic CYP2C subfamily of cytochrome P450 genes (*CYP2C9*, *CYPC19*, *CYP2D6*), in a total of 95 individuals. The large fraction of rare known risk variants and putative LOF variants identified in pharmacogenes important for metabolizing a broad range of drugs suggests that next-generation sequencing-based assays are warranted for comprehensive pharmacogenetic testing, as genotyping assays may miss or inaccurately detect such rare variants.

## Discussion

Here, we characterized clinically significant genetic variation in an ancestrally diverse Southeast Asian population and highlighted diversity in risk profiles for dominant and recessive genetic disorders, capturing the common disorders among Asians missed by prevailing screening panels. Although overall frequency of clinically actionable

**Table 2 | Identified alleles in pharmacogenes and the carrier frequency of associated actionable phenotypes among Singaporeans in the SG10K_Health cohort**

| Gene | Risk allele(s) | Phenotype | No. risk allele carriers/Total individuals genotyped (%) | | | |
|---|---|---|---|---|---|---|
| | | | CH | IND | MY | All |
| CACNA1S | decreased func.: rs772226819 | MHS | 0/5377 | 0/1891 | 1/1571 (0.1%) | 1/8839 (0.01%) |
| CFTR | rs115545701, rs202179988, rs78769542, rs74503330 | Responsive to ivacaftor | 9/5239 (0.2%) | 15/1796 (0.8%) | 7/1509 (0.5%) | 31/8544 (0.4%) |
| CYP2B6 | increased func.: *4, *22 | UM | 53/5452 (1.0%) | 12/1926 (0.6%) | 3/1594 (0.2%) | 68/8972 (0.8%) |
| | decreased func.: *6, *7, *9, *19, *26, *36 | RM | 596/5452 (10.9%) | 124/1926 (6.4%) | 68/1594 (4.3%) | 788/8972 (8.8%) |
| | no func.: *8, *12, *13, *18, *24 | IM | 1690/5452 (31.0%) | 819/1926 (42.5%) | 705/1594 (44.2%) | 3214/8972 (35.8%) |
| | | PM | 315/5452 (5.8%) | 320/1926 (16.6%) | 257/1594 (16.1%) | 892/8972 (9.9%) |
| CYP2C9 | decreased func.: *2, *8, *11, *14, *16, *29, *31, *37, *44, *50, *55 | IM | 408/5452 (7.5%) | 440/1926 (22.9%) | 119/1595 (7.5%) | 967/8973 (10.8%) |
| | no func.: *3, *13, *33, *39, *42, *45, *52 | PM | 14/5452 (0.3%) | 50/1926 (2.6%) | 5/1595 (0.3%) | 69/8973 (0.8%) |
| CYP2C19 | increased func.: *17 | UM/RM | 51/5452 (0.9%) | 280/1926 (14.5%) | 54/1595 (3.4%) | 385/8973 (4.3%) |
| | decreased func.: *10, *16, *26 | IM | 2434/5452 (44.6%) | 881/1926 (45.7%) | 619/1595 (38.8%) | 3934/8973 (43.8%) |
| | no func.:*2, *3, *4, *5, *6, *8, *24, *35 | PM | 765/5452 (14.0%) | 326/1926 (16.9%) | 140/1595 (8.8%) | 1231/8973 (13.7%) |
| CYP2D6 | increased func.: *1×2, *1×3, *2×2 | UM | 51/3870 (1.3%) | 39/1481 (2.6%) | 13/1124 (1.2%) | 103/6475 (1.6%) |
| | decreased func.: *9, *10, *10×2, *10×2+*83, *14, *17, *29, *41, *49, *49×2, *36-*10, *36×2-*10, *36×2-*10-*83, *36+*10×2+*83 | IM | 1667/3870 (43.1%) | 366/1481 (24.7%) | 413/1124 (36.7%) | 2446/6475 (37.8%) |
| | no func.: *3, *4, *4 N, *4+*4 N, *4+*68, *5, *6, *7, *13, *15, *21, *21×2, *31, *36, *36×2, *36×3, *68, *69, *99, *101, *114 | PM | 110/3870 (2.8%) | 55/1481 (3.7%) | 66/1124 (5.9%) | 231/6475 (3.6%) |
| CYP3A4 | decreased func.: *22 | Decreased metabolism | 0/5452 | 18/1926 (0.9%) | 6/1595 (0.4%) | 24/8973 (0.3%) |
| CYP3A5 | normal func.: *1 | NM | 534/5448 (9.8%) | 291/1917 (15.2%) | 251/1562 (16.1%) | 1076/8927 (12.1%) |
| | no func.: *3, *6, *7 | IM | 2029/5448 (37.2%) | 765/1917 (39.9%) | 706/1562 (45.2%) | 3500/8927 (39.2%) |
| | | PM | 2688/5448 (49.3%) | 844/1917 (44.0%) | 589/1562 (37.7%) | 4121/8927 (46.2%) |
| CYP4F2 | decreased func.: rs2108622 | Increased warfarin dose requirement | 2178/5283 (41.2%) | 1252/1839 (68.1%) | 531/1548 (34.3%) | 3961/8670 (45.7%) |
| DPYD | decreased func.: HapB3, rs186169810, rs112766203 | IM | 39/5445 (0.7%) | 82/1915 (4.3%) | 4/1562 (0.3%) | 125/8922 (1.4%) |
| | no func.: *2, *8, rs72549304, rs59086055, rs138616379, rs55674432, rs72549308, rs3918290 | PM | 0/5445 | 0/1915 | 1/1562 (0.1%) | 1/8922 (0.01%) |
| F5 | rs6025 | Increased risk of VTE | 2/5378 (0.04%) | 44/1872 (2.4%) | 6/1565 (0.4%) | 52/8815 (0.6%) |
| G6PD | WHO Class II, III deficient alleles [a] | Deficient | 53/5502 (1.0%) | 18/1941 (0.9%) | 16/1608 (1.0%) | 87/9051 (1.0%) |
| | | Variable | 179/5502 (3.3%) | 34/1941 (1.8%) | 44/1608 (2.7%) | 257/9051 (2.8%) |
| HLA-A | deficient: A*31:01:02 | Increased risk of SCAR | 147/5468 (2.7%) | 99/1919 (5.2%) | 11/1596 (0.7%) | 257/8983 (2.9%) |
| HLA-B | deficient: B*15:02:01, B*58:01 | Increased risk of SCAR | 1452/5468 (26.6%) | 200/1918 (10.4%) | 441/1596 (27.6%) | 2093/8982 (23.3%) |
| | deficient: B*57:01:01 | Abacavir hypersensitivity | 33/5468 (0.6%) | 265/1918 (13.8%) | 28/1596 (1.8%) | 326/8982 (3.6%) |
| IFNL3 | decreased response: rs12979860, rs8099917 | Decreased response to peginterferon alfa-2a/2b | 620/5168 (12.0%) | 750/1801 (41.6%) | 197/1520 (13.0%) | 1567/8489 (18.5%) |
| IFNL4 | decreased response: rs12979860, rs11322783 | Decreased response to peginterferon alfa-2a/2b | 603/5079 (11.9%) | 748/1768 (42.3%) | 198/1521 (13.0%) | 1549/8368 (18.5%) |
| NAT2 | rapid allele: *4 | Rapid acetylator | 1452/5452 (26.6%) | 119/1925 (6.2%) | 248/1595 (15.6%) | 1819/8972 (20.3%) |
| | slow allele: *5, *6, *6 A, *7, *7B | Intermediate acetylator | 2252/5452 (41.3%) | 340/1925 (17.7%) | 537/1595 (33.7%) | 3129/8972 (34.9%) |
| | | Slow acetylator | 1015/5452 (18.6%) | 404/1925 (21.0%) | 390/1595 (24.5%) | 1809/8972 (20.2%) |
| NUDT15 | no func.: rs116855232 | IM | 1037/5362 (19.3%) | 265/1883 (14.1%) | 183/1560 (11.7%) | 1485/8805 (16.9%) |
| | | PM | 52/5362 (1.0%) | 13/1883 (0.7%) | 13/1560 (0.8%) | 78/8805 (0.9%) |
| RYR1 | increased func.: rs112563513, rs121918593, rs18192168 | MHS | 4/4947 (0.1%) | 1/1727 (0.1%) | 0/1486 | 5/8160 (0.1%) |
| SLCO1B1 | decreased func.: rs4149056 | | 1092/5366 (20.4%) | 268/1896 (14.1%) | 250/1575 (15.9%) | 1610/8837 (18.2%) |

**Table 2 (continued)**

| Gene | Risk allele(s) | Phenotype | No. risk allele carriers/Total individuals genotyped (%) | | | |
|---|---|---|---|---|---|---|
| | | | CH | IND | MY | All |
| | | Intermediate risk of statin-related myopathy | 69/5366 (1.3%) | 2/1896 (0.1%) | 11/1575 (0.7%) | 82/8837 (0.9%) |
| | | Increased risk of statin-related myopathy | | | | |
| *TPMT* | no func.: *3, *3 A, *3 C, *29 | IM | 124/4297 (2.9%) | 32/1245 (2.6%) | 52/1206 (4.3%) | 208/6748 (3.1%) |
| | | PM | 0/4297 | 1/1245 (0.1%) | 3/1206 (0.3%) | 4/6748 (0.1%) |
| *UGT1A1* | increased func.: *3 | IM | 1737/4341 (40.0%) | 809/1492 (54.2%) | 332/1201 (27.6%) | 2878/7034 (40.9%) |
| | decreased func.: *6, *28, *37, *80 + *28, *80 + *37 | PM | 140/4341 (3.2%) | 259/1492 (17.4%) | 16/1201 (1.3%) | 415/7034 (5.9%) |
| *VKORC1* | rs9923231, rs7294, rs2359612, rs9934438 | Higher warfarin sensitivity | 4569/4630 (98.7%) | 428/1507 (28.4%) | 1282/1348 (95.1%) | 6279/7485 (83.9%) |

[a]G6PD risk alleles: (WHO Class II) Canton, Chatham, CoimbraShunde, Kaiping, Maewo, Mediterranean, Namouru, Nankang, Nilgiri, Valladolid, Viangchan_Jammu; (WHO Class III) Asahi, Chinese-5, Gaohe, Hammersmith, Kalyan-Kerala, Mahidol, Montalbano, Nanning, Orissa, QuingYan, UbeKonan.
Abbreviations
CH Chinese, IND Indian, MY Malay, IM intermediate metabolizer, NM normal metabolizer, PM poor metabolizer, RM rapid metabolizer, UM ultrarapid metabolizer, func. function, MHS Malignant hyperthermia susceptibility SCAR Severe Cutaneous Adverse Reactions, VTE venous thromboembolism.

SFs was comparable to European-centric cohorts, there were differences in concentration of disease burden, exemplified by the higher risk for FH among Chinese in contrast to the higher risk for HBOC among European-descent populations[6,41]. Our data also showed that disease risk and carrier burden were varied even among Asian ancestry groups, driven by distinctive prevalence of ancestry-specific recurrent variants. In this study, we characterized genetic risk in Malays, a severely under-represented Austronesian-speaking Southeast Asian population, and highlighted distinction in their disease risk profiles compared to East and South Asians.

Emblematic of current Eurocentric genomic medicine guidelines, we found 27% of severe recessive disorder genes with carrier frequencies exceeding 1-in-200 Asians are unrepresented in ACMG carrier screening recommendations or commercial carrier screening panels. Left unaddressed, Asian couples will be at greater risk for conditions missed by existing screening panels and based on our lower-bound estimate of 0.51% Singaporean ARCs for severe recessive disorders, conservative projection to a combined reproductive-age population of 94 million encompassing South India, South China and Austronesian-speaking Southeast Asia would translate to almost half a million at-risk Asian couples standing to benefit from carrier screening. This is slightly lower compared to the ARC rate of 0.8%-1.0% observed in a population of Estonian and Dutch couples of European ancestry[42] and is likely due to the under-reporting of Asian variants in clinical databases and literature[43,44]. Our findings underscore the importance of diverse representation in genetic risk profiling across disease domains and in development of clinical recommendations, particularly within multi-ethnic settings, to address disparities in health care delivery and outcomes.

Cross-ancestry differences extend beyond disease prevalence to the spectrum of genetic variants for the same gene, potentially accounting for inter-population variability in disease manifestation. For instance, the *GJB2* c.35delG (p.Gly12Valfs*2) variant associated with profound hearing loss is prevalent among populations of European-descent[22] but rare among Asians of Chinese and Malay ancestry, most of whom harbour the Val37Ile variant associated with mild-to-moderate hearing impairment. Notably, whereas cystic fibrosis is prevalent in European-descent populations and frequently associated with *CFTR* c.1521_1523delCTT (p.Phe508del) variant, this is rare in Asia where *CFTR*-related CBAVD and pancreatitis are more frequently observed together with *CFTR* c.1210-11T > G and Gln1352His variants[45,46]. Under-recognition of such genotype-phenotype associations can have consequences, as symptoms for less-characterized disorders afflicting non-European groups may go undetected and result in misdiagnosis or missed opportunities for early intervention.

The prevalence of cryptic admixture in our multi-ethnic cohort highlights the pitfalls of over-reliance on self-reported R/E for genetic risk profiling[5,12]. Notably, we observed a self-identified Chinese adult female carrying a Malay founder variant for *BRCA1* (Asn909Lysfs*6)[20,47] as well as numerous Chinese and Indian individuals harbouring variants identified recurrently among Malays (e.g. *ABCA4* Arg24His, *ARHGEF18* c.826-1G > A); all of whose genetic ancestry includes an appreciable (10%-20%) Malay ancestral component. This is consistent with Singapore's history of immigration, epitomized by admixture among the Peranakan community established through inter-marriage between Chinese and Indian immigrants with native Malays since the 15th century[48]. Our findings highlight that genetic susceptibility to health disorders cuts across ethnic boundaries, especially as populations become increasingly admixed worldwide, driven by intercontinental unions and human migration accelerated by socio-geopolitical factors. With Asians accounting for the rapid rise in minority/immigrant groups in the United States and Europe[49,50], integration of Asian population-derived data will be increasingly relevant for more precise clinical risk assessment and narrowing gaps in health care delivery. At

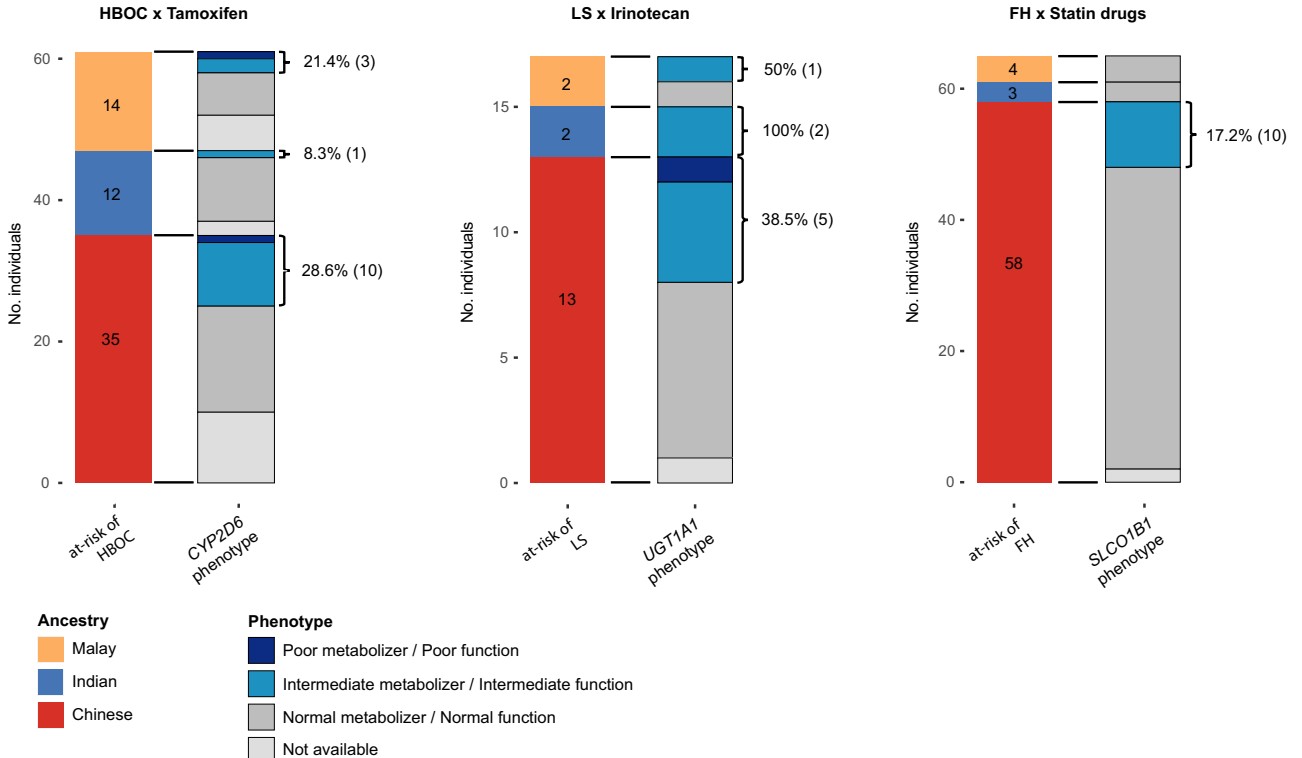

**Fig. 3 | Carriers of both germline pathogenic/likely pathogenic (P/LP) variant in a CDC Tier 1 condition and pharmacogenetic variant associated with an actionable phenotype to drugs used for treatment.** Only pharmacogenetic variant-drug combinations supported by PharmGKB Level 1A/1B evidence were considered. HBOC hereditary breast and ovarian cancer syndrome, LS Lynch syndrome, FH familial hypercholesterolemia, *n* number of germline P/LP variant carriers for the indicated CDC Tier 1 genetic conditions among 9051 Singaporeans.

present, the 'informational disparity' stemming from Eurocentric studies[11] limits clinical interpretation of variants detected in underrepresented ancestry groups and as indicated by our data, there are Asian-specific pathogenic variants that are currently classified as VUS, which can be reclassified with increased detection through widespread testing.

Our study comprehensively profiled high confidence gene-drug interactions across three Asian ancestries, using whole-genome sequencing to uniformly analyse the pharmacogenomics of a large cohort of these ancestries. We demonstrated contrasting drug response profiles along ancestry lines driven by variability in allele frequencies, consistent with a smaller Singaporean study[51], contributing to distinctive pharmacologic susceptibility across ancestry groups. Importantly, we showed that approximately one-fifth of individuals with predisposition to a genetic disorder are at risk of therapeutic failure or life-threatening toxicity for drugs commonly prescribed to treat the disease. This highlights that a substantial fraction of genetically susceptible individuals could benefit from pre-emptive pharmacogenomics to optimize their therapeutic treatments and avoid severe toxicities, indicating opportunities to forge a more comprehensive clinical care by combining pharmacogenomics and genetic disease testing.

This work demonstrates that Asians are a diverse population with complex genomic architecture and extensive genetic variability. Although a conservative estimate of Asian population genetic risk given the focus on known disease genes and coding variants, our data provides opportunities to address disparities in existing knowledge by demonstrating the contrast in risk profiles of monogenic disorders between European and Asian ancestry groups and the need for expanded carrier testing among Asians. Beyond diversity, we also showed that monogenic disorder pathogenic variants are mostly rare, with >85% carried in only 1–2 individuals, supporting the need for comprehensive sequence-based testing as opposed to array-based single nucleotide polymorphism (SNP) genotyping[52]. Critically, we highlighted the prevalence of cryptic admixture and limitation of self-reported R/E in estimating genetic risk burden in an ethnically diverse population and demonstrated the potential benefit of coupling pharmacogenomics with clinical genetic testing. As genomic profiling gains traction in mainstream precision medicine, the diversified representation of all population groups in genomic research will be imperative to level the gaps in health disparity for a truly equitable delivery of precision medicine.

## Methods
### Study population
The source dataset used for this study was derived from the SG10K_Health project. Individuals from six participating studies (Supplementary Table 1) were recruited with signed informed consent from the participating individual or parent/guardian in the case of minors. Germline DNA for whole genome sequencing were extracted from whole blood or cord blood (for birth cohort) specimens of enrolled individuals according to respective study protocols. All studies were approved by relevant institutional ethics review board detailed in Supplementary Table 1. The final analysed cohort comprised 9051 individuals inferred to be unrelated to the second degree through kinship analysis, with global genetic ancestry (henceforth, 'genetic ancestry') inferred through admixture analysis (subsection: Kinship and admixture inference). For ancestry analysis, self-reported race/ethnicity (R/E) was captured from the respective national identification document of participating individuals.

### Sequencing and bioinformatics analysis
We performed whole genome sequencing for germline DNA on Illumina Hiseq X platform to a target depth of 30X or 15X. Resulting

paired-end sequencing reads were jointly-processed in a standardized bioinformatics pipeline that involved alignment to the human reference genome (hg38) using Burrows-Wheeler Aligner (BWA-MEM, v0.7.17)[53] followed by GenomeAnalysisToolKit (GATK, v4.0.6.0) best practices workflow to produce a jointly-genotyped variant call file (VCF) comprising 9,770 samples[54,55]. To accelerate variant annotation, we trimmed the full VCF to retain only positions overlapping our genes list (subsection: Gene selection) and samples that were unrelated up to the seconds degree (n = 9,051). Heterozygous sites were re-genotyped to "no call" status if the following criteria were unmet: (1) allele balance between 20% and 80%, (2) minimum read depth of 5, (3) minimum genotype quality of 20. We performed variant annotation using Ensembl Variant Effect Predictor (VEP, release 100.0)[56] to include information such as overlapping genes, consequence type, Human Genome Variation Society (HGVS) nomenclature for DNA and protein alterations, population allele frequencies and in silico pathogenicity prediction scores from REVEL (rare exome variant ensemble learner)[57], PrimateAI[58] and SpliceAI[59]. As VEP provides one predicted consequence for each transcript, we selected the consequence on the MANE (Matched Annotation from NCBI and EMBL-EBI) transcript. Where a gene does not have a MANE transcript, the transcript with the most deleterious variant consequence and/or the longest gene transcript affected was selected. Samples sequenced to target depth of 30X versus 15X were evaluated for potential batch effects and the carrier frequencies of identified variants were shown to be strongly correlated (Pearson's r > 0.86, Supplementary Fig. 5).

### Gene selection

We consolidated a list of 4143 genes (Supplementary Data 9) associated with autosomal dominant (AD), autosomal recessive (AR), and X-linked monogenic disorders from three sources: (1) 3252 genes with diagnostic-grade (green) status from PanelApp[60] (accessed 5 May 2020), (2) 5,506 genes from Online Mendelian Inheritance in Man (OMIM) (www.omim.org, accessed 21 May 2020), (3) 4121 genes from in-house gene panels for cardiomyopathies, cancer predisposition, paediatrics and ophthalmology. We excluded genes linked to repeat expansion disorders.

### Identification of loss-of-function intolerant (LOFi) genes.

We defined a total of 1,856 genes as LOFi if any one of the following criteria was fulfilled: (1) genes considered to be haploinsufficient by the Clinical Genome Resource[61] (ClinGen, n = 727, accessed May 01 2020), (2) genes with ≥3 variants classified as pathogenic or likely pathogenic in ClinVar[62] with a review status of at least 2 gold stars (i.e. is a practice guideline, or has been reviewed by expert panel, or has multiple submitters with criteria provided and no conflicts; subsequently referred to as 'ClinVar TwoPlus' variants) and were one of the following variant types: frameshift insertion/deletion, nonsense, essential splice site variant (±2 residues from splice site) (n = 587, accessed September 09, 2020), (3) genes with ExAC pLI[63] (probability of being LOFi, obtained from dbNSFP 4.0) score > 0.9 (n = 983).

### Variant classification and interpretation

We retained variants that overlapped genes in our consolidated gene list for curation if reported in ClinVar or had a SG10K_Health allele frequency <0.05, and were categorised into one of the following groups (Supplementary Fig. 1): (1) Pathogenic/Likely Pathogenic (P/LP), (2) Variants of uncertain significance-favour pathogenic (VUS-FP), (3) Variants of uncertain significance (VUS), (4) Unclassified.

### Pathogenic/Likely Pathogenic (P/LP).

We further subset variants in this group into three categories: (1) Tier1A_TwoPlus: ClinVar TwoPlus variants were considered high confidence known pathogenic variants and automatically classified as P/LP. Novel single nucleotide variants that result in known amino acid codon change that has a ClinVar TwoPlus status were also categorized as P/LP. (2)Tier1A_Conflicting: Variants in ClinVar with conflicting interpretations but ≥4P/LP submissions were considered P/LP whereas those with 1-3P/LP submissions were manually curated according to American College of Medical Genetics and Genomics and the Association for Molecular Pathology (ACMG/AMP) guidelines[64], taking into consideration allele frequency, in silico scores and reports in literature (Human Gene Mutation Database (HGMD)[65] and PubMed). Known variants that occurred in cis such as GAA c.752C > T;c.761 C > T were counted as one event. (3) Tier1B: LOF variants (frameshift insertions/deletions, nonsense, essential splice site variants) that were either absent in ClinVar or that were in ClinVar but did not meet our preceding criteria, were manually curated according to the ACMG/AMP PVS1 criterion using the high-throughput, automated application AutoPVS1 (v1.1)[66]. Variants fulfilling the following criteria were considered P/LP: (a) LOF consequence in MANE transcript; for genes without MANE transcript, ClinVar and the National Center for Biotechnology Information (NCBI) were referenced to determine the LOF variant affected a clinically relevant transcript, and (b) AutoPVS1 indicated PVS1 strength of "Very Strong", or (c) there are ≥2 ClinVar TwoPlus P/LP variants located downstream of the variant. Truncating variants in TTN were separately assessed using CardioClassifier[67] (v.0.2.0) for P/LP classification.

### Variants of uncertain significance (VUS) and VUS-favour pathogenic (VUS-FP).

Variants that did not meet our P/LP criteria were considered VUS. We also considered the following variants as VUS: (1) Variants in ClinVar with conflicting interpretations but ≥4 VUS submissions, (2) LOF variants in close proximity, which upon manual inspection using Integrative Genomics Viewer (IGV, v2.8.2)[68] showed a non-frameshift insertion/deletion consequence. We defined VUS with potential LOF consequence as VUS-FP if the following criteria were met: (a) missense variants with REVEL score >0.7 and are located in a 'hotspot' (defined as a rolling window of 25 bp with >2 ClinVar TwoPlus P/LP variants and with the number of benign/likely benign variants less than ClinVar TwoPlus P/LP variants), or (b) cryptic splice variants with SpliceAI maximum score >0.8 and occurred in genes with ≥5 ClinVar TwoPlus P/LP LOF (nonsense/frameshift/canonical splice) variants. All remaining variants that did not meet any of the P/LP, VUS or VUS-FP criteria were categorised as "Unclassified".

### Gross deletions.

We derived gross deletions included for our analysis from a structural variant (SV) callset generated by the SG10K_SV workgroup. For each sample, CRAM file was processed using Manta (v1.6)[69] to identify candidate SVs. Subsequently, the SVs across all samples were merged using svimmer (v0.1), and then re-genotyped using graphtyper2 (v2.5.1)[70]. To identify high-confidence CNVs, duphold (v0.2.3)[71] was performed to add read-depth information to the SV calls. We considered only deletions that overlapped at least an exon of the MANE transcript in our LOFi gene list and that met the following criteria: (1) length of 500 bp–10 Mbp, (2) deletions with duphold DHFC, DHFFC, DHBFC values <0.7 and DHSP > 1. We visually confirmed candidate CNVs using samplot (v1.0.20)[72]. We separately identified deletions in SMN1 using SMNCopyNumberCaller (v1.1.1)[73] only on samples with 30X sequencing coverage.

### Virtual mating simulation

To estimate frequency of at-risk couples (ARCs) for recessive disorders, we considered all possible matings within each ancestry group, regardless of sex[42] (Chinese, CH = 15,133,251; Indian, IND = 1,882,770; Malay, MY = 1,292,028). We considered a simulated couple to be at-risk if both carried P/LP variants in one or more AR genes associated with severe recessive disorder[32]. We created an exclusion list comprising variants considered to cause clinically significant disease only in trans with a more severe P/LP variant, hence if a theoretical couple were simulated to have an offspring that is homozygous for a variant in the

exclusion list or compound heterozygous for two variants within the exclusion list, the couple was not considered to be an ARC.

## Kinship and admixture inference

To perform kinship analysis, we extracted a set of known polymorphic sites from the full VCF using Somalier (v0.2.13)[74] and processed using PLINK (v1.90b3.46)[75] to produce a PLINK BED reference panel, consisting single nucleotide polymorphisms (SNPs) pruned with $r^2 > 0.5$ (using PLINK recommended settings of window sizes of 50 SNPs with steps of 5 SNPs across the genome). We used Kinship-based Inference for Genome-wide association studies (KING, ver 2.2.3)[76] to calculate pairwise kinship coefficients and considered pairs of samples with kinship coefficient ≥0.0884 as related, and randomly select one from each pair for exclusion.

For global ancestry inference, we performed admixture analysis to estimate the proportions of three hypothetical ancestral components in each sample on ADMIXTURE (ver 1.3.0)[15] with K = 3 using the same PLINK BED reference panel. The hypothetical components of K = 3 has been demonstrated to sufficiently delineate the three major ancestry groups (Chinese, Indian, Malay) in a Singaporean cohort[14]. The highest of the three estimated ancestral components for each individual was inferred as genetic ancestry. For the purpose of our analyses, "genetic ancestry" assigned to each individual is a statistical construct calculated from inherited genetic variants and is not equivalent to, nor intended to replace, self-reported race or ethnicity, which are social constructs identified by the individuals.

To estimate local ancestry, we used phased genotypes generated using EAGLE (v2.4.1) and retained only SNPs with minor allele frequency ≥1% and call rate of ≥ 0.5. We selected 100 individuals from each ancestry group with the highest respective ancestral component, and the combined 300 individuals representing Chinese, Indian and Malay ancestry groups were used as the reference panel for inference of local ancestry using RFMix (v2.03-r0)[77] on default settings. In the analysis of discordant variant carriers in Fig. 2b, we defined ancestry-specific variants by the following criteria: (1) P/LP variants with allele count ≥5, and (2) the variant exclusively occurs in an allele with the same inferred local ancestry. For instance, a Chinese-specific variant is one that occurs exclusively in alleles with inferred local ancestry of Chinese origin.

## Pharmacogenomic variants

For profiling the pharmacogenomic landscape, we consolidated a list of 23 pharmacogenes from the CPIC (Clinical Pharmacogenetics Implementation Consortium) drug-gene pair list (Supplementary Data 10, accessed Aug 30 2021) with Pharmacogenomics Knowledgebase (PharmGKB) clinical annotation level of evidence 1A/1B, which are defined as: (Level 1A) gene-drug combinations with variant-specific prescribing guidance in existing clinical guideline annotation or an FDA-approved drug label annotation, and minimally one publication supporting the clinical annotation, or (Level 1B) gene-drug combinations with no variant-specific prescribing guidance but has a high level of evidence supporting the association with at least two independent publications[78]. Referencing the CPIC and Pharmacogene Variation Consortium (PharmVar) repositories (accessed April 2021), we identified known pharmacogenetic alleles of these 23 genes using the following methods: (a) Cyrius (v1.0)[79] (CYP2D6) and Aldy (v3.1)[80] for genes with star allele nomenclature, (b) HLA-HD (v1.3.0)[81] for HLA-A and HLA-B alleles, (c) VCF-derived for genes with pharmacogenetic alleles defined by dbSNP rsIDs. Allele frequencies for each allele with a functional status associated with known pharmacogenetic phenotype is tabulated in Supplementary Data 6, whereas the carrier frequency of actionable pharmacogenetic phenotypes associated with the 23 pharmacogenes is tabulated in Table 2, and further consolidated by actionable phenotype with therapeutic recommendation guidelines in Supplementary Data 7. Carrier frequency for diplotypes associated

with actionable phenotypes for pharmacogenes with star nomenclature is consolidated in Supplementary Data 11.

For identification of potentially deleterious novel variants (i.e. not found in CPIC or PharmVar), we filtered for putative LOF variants (frameshift insertions/deletions, nonsense, essential splice site) that: (a) are located in MANE transcript, and (b) AutoPVS1 indicated PVS1 strength of "Very Strong", and (c) occurred in 10 of the 23 pharmacogenes in our list, for which LOF is a mechanism associated with the actionable phenotype (CYP2B6, CYP2C9, CYP2C19, CYP2D6, DPYD, G6PD, NUDT15, SLCO1B1, TPMT, UGT1A1). Upon manual review, one variant (SLCO1B1 c.1738C > T (p.Arg580*), rs71581941) was removed due to poor read coverage.

## Statistical analysis

We performed all statistical analyses using R version 4.1.0[82]. Cohort data, gene- and variant-level carrier frequencies were tabulated with descriptive statistics. We performed two-sided Fisher's exact test for comparison of proportions for categorical variables, whereas two-sided Wilcoxon rank-sum test was used for comparing continuous variables. p values were adjusted with Benjamini-Hochberg correction for multiple testing. Binomial logistic regression was used for comparison of LDL cholesterol levels against LDLR variant status, correcting for age, sex, ancestry and lipid-lowering medication intake.

## Reporting summary

Further information on research design is available in the Nature Research Reporting Summary linked to this article.

## Data availability

Data (WGS and intermediate files) for all analyses and regeneration of all display items contain individual-level data including genotypes, and is made available to researchers registered through the SG10K_Health Data Access Portal (https://www.npm.sg/collaborate/partners/sg10k/). Requestors should be bona fide researchers and are required to submit a Data Access Request outlining the proposed research for approval by the Data Access Committee, which convenes monthly. Data for this study were obtained under Data Access Application NPM00003.

## Code availability

All code to perform all analyses and regenerate all the figures in this manuscript is provided at https://github.com/csockhoai/SG10KMed and released at https://doi.org/10.5281/zenodo.7057754[83].

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

## Acknowledgements

This study made use of data collected from the following participating cohorts in Singapore: (1) The Health for Life in Singapore (HELIOS) study at the Lee Kong Chian School of Medicine, Nanyang Technological University, Singapore, (2) The Growing Up in Singapore Towards Healthy Outcomes (GUSTO) study jointly hosted by the National University Hospital (NUH), KK Women's and Children's Hospital (KKH), the National University of Singapore (NUS) and the Singapore Institute for Clinical Sciences (SICS), Agency for Science Technology and Research (A*STAR), (3) The Singapore Epidemiology of Eye Diseases (SEED) cohort at Singapore Eye Research Institute (SERI), (4) The Multi-Ethnic Cohort (MEC), (5) The SingHealth Duke-NUS Institute of Precision Medicine (PRISM) cohort, (6) The Tan Tock Seng Hospital (TTSH) Personalised Medicine Normal Controls cohort. The views expressed are those of the author(s) are not necessarily those of the National Precision Medicine investigators, or institutional partners. We thank all investigators, staff members and study participants who made the National Precision Medicine Programme possible. The computation for this study was partially performed on resources of the National Supercomputing Centre, Singapore (https://www.ncss.sg). We also thank Jack Ow, Shimin Ang, Rodrigo Toro, Pauline Chen, Chih Chuan Shih, Zheng Li, Lorenz Gerber, Wing Cheong Wong, Dimitar Kenanov, Ashar Jamil Malik, Chandra Verma for bioinformatics support. This study made use of data generated as part of the Singapore National Precision Medicine program funded by the Industry Alignment Fund (Pre-Positioning) (IAF-PP: H17/01/a0/007). The participating study cohorts were supported by the following funding sources: (1) HELIOS study by grants from a Strategic Initiative at Lee Kong Chian School of Medicine, the Singapore Ministry of Health (MOH) under its Singapore Translational Research Investigator Award (NMRC/STaR/0028/2017) and the IAF-PP: H18/01/a0/016, (2) GUSTO study by the Singapore National Research Foundation under its Translational and Clinical Research (TCR) Flagship Programme and administered by the Singapore MOH's National Medical Research Council (NMRC) Singapore (NMRC/TCR/004-NUS/2008, NMRC/TCR/012-NUHS/2014) with additional funding provided by SICS and IAF-PP: H17/01/a0/005, (3) SEED study by NMRC/CIRG/1417/2015, NMRC/CIRG/1488/2018, NMRC/OFLCG/004/2018, (4) MEC study by NMRC grant 0838/2004, BMRC grant 03/1/27/18/216, 05/1/21/19/425, 11/1/21/19/678 to NUS and National University Health System (NUHS) Singapore, (5) PRISM cohort study by NMRC/CG/M006/2017_NHCS, NMRC/STaR/0011/2012, NMRC/STaR/0026/2015, EYE ACP-PRISM PRECISION MEDICINE INITIATIVE FUND 05/FY2020/EX/06-A41, Lee Foundation and Tanoto Foundation, (6) TTSH cohort study by NMRC/CG12AUG2017 and CGAug16M012. Additional funding support includes grants under National Research Foundation Singapore administered by the Singapore Ministry of Health's National Medical Research Council to the following individuals: National Precision Medicine Programme (NPM) PHASE II

FUNDING (MOH-000588) to W.K.L., National Medical Research Council Singapore Clinician-Scientist Award (NMRC/CSA-INV/0017/2017, MOH-000654) to J.N., National Medical Research Council Singapore Clinician-Scientist Award (CSAINV21Jun-0003) to S.S.J., and Clinician-Scientist Award Senior Investigator (NMRC/CSA-SI/0012/2017) to C-Y.C; as well as funding from Agency for Science, Technology, and Research (A*STAR) of Singapore to J. Liu. The funders had no role in study design, data collection, data analysis, data interpretation, writing of the manuscript or decision to submit for publication.

## Author contributions

Data were generated, curated, and analysed by S.H.C, Y.B., P.T., S.S.J, J.N., W.K.L., J.X.T., J.L.K., N.B., M.G-P., M.H., R.T-M., J.H.J.T., J.J., Z.L., J.F.C., Y.S.C., S.D., L.L.G., E.S.L., E.W., T.Y.W., S.P., J. Liu, C-Y.C., B.E., N.K., K.P.L., X.S., K.K.Y., J.C.C., T.A., K.H.K.B., C.B., M.Li C., M.Ling C., W.J.C., C.W.L.C., R. Dalan, R. Dorajoo, P.E., J.G.E., P.D.G., I.K., L.N.L., J. Lee, Y.S.L., H.L., C.W.L., T.H.L., M.L., S.M-S., T.H.M., S.Q.M., H.K.N., C.H.P., E.R., T.H.R., C.S., E.T., W.C.S., D.T., Y.Y.T., Y-C.T., L.G.T., P.K.T., R.M.v.D., L.V., G.W.K., A.W., C.Y., F.Y., Y.W.Y. Data were interpreted by S.H.C., Y.B., P.T., S.S.J., J.N., W.K.L., E-S.T., S.A.C., C.L.D., R.F., D.G., D.L.M.G., K.J., S.K., C.G.L., S-C.L., T.S., E.S.T., E.K.T. Study was designed and manuscript written by S.H.C, Y.B., P.T., S.S.J, J.N., W.K.L. Study was jointly supervised by P.T., S.S.J, J.N., W.K.L.

## Competing interests

The authors declare no competing interests.

## Additional information

Sock Hoai Chan [1,2,3,68], Yasmin Bylstra [4,68], Jing Xian Teo [4], Jyn Ling Kuan [4], Nicolas Bertin [5], Mar Gonzalez-Porta [5], Maxime Hebrard [5], Roberto Tirado-Magallanes [5], Joanna Hui Juan Tan [5], Justin Jeyakani [5], Zhihui Li [5], Jin Fang Chai [6], Yap Seng Chong [7,8], Sonia Davila [4,9,10], Liuh Ling Goh [11], Eng Sing Lee [3,12], Eleanor Wong [13], Tien Yin Wong [14], SG10K_Health Consortium*, Shyam Prabhakar [15], Jianjun Liu [16,17], Ching-Yu Cheng [14,18], Birgit Eisenhaber [13,19], Neerja Karnani [20,21,22], Khai Pang Leong [11,23], Xueling Sim [6], Khung Keong Yeo [4,24,25], John C. Chambers [3,26,27], E-Shyong Tai [6,26,17,25], Patrick Tan [4,13,26,28,29] ✉, Saumya S. Jamuar [4,10,30,31] ✉, Joanne Ngeow [1,2,3,32] ✉ & Weng Khong Lim [4,10,28] ✉

[1]Cancer Genetics Service, Division of Medical Oncology, National Cancer Centre Singapore, Singapore 169610, Singapore. [2]Oncology Academic Clinical Program, Duke-NUS Medical School, Singapore 169857, Singapore. [3]Lee Kong Chian School of Medicine, Nanyang Technological University, Singapore 308232, Singapore. [4]SingHealth Duke-NUS Institute of Precision Medicine, Singapore 169609, Singapore. [5]Genome Research Informatics & Data Science Platform, Genome Institute of Singapore, Agency for Science, Technology and Research, Singapore 138672, Singapore. [6]Saw Swee Hock School of Public Health, National University of Singapore, Singapore 117549, Singapore. [7]Department of Obstetrics & Gynaecology, Yong Loo Lin School of Medicine, National University of Singapore, Singapore 119228, Singapore. [8]Singapore Institute for Clinical Sciences, Singapore 117609, Singapore. [9]Cardiovascular and Metabolic Disorders Program, Duke-NUS Medical School, Singapore 169857, Singapore. [10]SingHealth Duke-NUS Genomic Medicine Centre, Singapore 168582, Singapore. [11]Personalized Medicine Service, Tan Tock Seng Hospital, Singapore 308433, Singapore. [12]National Healthcare Group Polyclinics, Singapore 138543, Singapore. [13]Genome Institute of Singapore, Agency for Science, Technology and Research, Singapore 138672, Singapore. [14]Singapore Eye Research Institute, Singapore National Eye Centre, Singapore 168751, Singapore. [15]Laboratory of Systems Biology and Data Analytics, Genome Institute of Singapore, Agency for Science, Technology and Research, Singapore 138672, Singapore. [16]Human Genomics, Genome Institute of Singapore, Agency for Science, Technology and Research, Singapore 138672, Singapore. [17]Department of Medicine, Yong Loo Lin School of Medicine, National University of Singapore, Singapore 119228, Singapore. [18]Ophthalmology & Visual Sciences Academic Clinical Program (Eye ACP), Duke-NUS Medical School, Singapore 169857, Singapore. [19]Bioinformatics Institute, Agency for Science, Technology and Research, Singapore 138671, Singapore. [20]Human Development, Singapore Institute for Clinical Sciences, Singapore 117609, Singapore. [21]Clinical Data Engagement, Bioinformatics Institute, Agency for Science, Technology and Research, Singapore 138671, Singapore. [22]Department of Biochemistry, Yong Loo Lin School of Medicine, National University of Singapore, Singapore 117596, Singapore. [23]Department of Rheumatology, Allergy and Immunology, Tan Tock Seng Hospital, Singapore 308433, Singapore. [24]Department of Cardiology, National Heart Centre Singapore, Singapore 169609, Singapore. [25]Duke-NUS Medical School, Singapore 169857, Singapore. [26]Precision Health Research Singapore (PRECISE), Singapore 139234, Singapore. [27]Department of Epidemiology and Biostatistics, Imperial College London, London W2 1PG, UK. [28]Cancer & Stem Cell Biology Program, Duke-NUS Medical School, Singapore 169857, Singapore. [29]Cancer Science Institute of Singapore, National University of

Singapore, Singapore 117599, Singapore. [30]Genetics Service, Department of Paediatrics, KK Women's and Children's Hospital, Singapore 229899, Singapore. [31]Paediatric Academic Clinical Program, Duke-NUS Medical School, Singapore 169857, Singapore. [32]Institute of Molecular and Cellular Biology, Agency for Science, Technology and Research, Singapore 138673, Singapore. [68]These authors contributed equally: Sock Hoai Chan, Yasmin Bylstra. *A list of authors and their affiliations appears at the end of the paper. ✉e-mail: patrick.tan@precise.cris.sg; Saumya.S.Jamuar@singhealth.com.sg; joanne.ngeow@ntu.edu.sg; wengkhong.lim@duke-nus.edu.sg

## SG10K_Health Consortium

Tin Aung[18,14], Kenneth Hon Kim Ban[22,33], Claire Bellis[16,34], Miao Li Chee[14], Miao Ling Chee[14], Wen Jie Chew[35], Calvin Woon-Loong Chin[24,36], Stuart A. Cook[9,37,38], Rinkoo Dalan[39,40], Rajkumar Dorajoo[16,41], Chester L. Drum[42,17], Paul Elliott[43], Johan G. Eriksson[7,8,44,45], Roger Foo[13,46], Daphne Gardner[47], Peter D. Gluckman[8], Denise Li Meng Goh[48], Kanika Jain[4,49], Sylvia Kam[10,30], Irfahan Kassam[50], Lakshmi Narayanan Lakshmanan[50], Caroline G. Lee[22,28], Jimmy Lee[3,51], Soo-Chin Lee[52,29], Yung Seng Lee[20,53], Hengtong Li[14], Chia Wei Lim[11], Tock Han Lim[54], Marie Loh[3,27,55], Sebastian Maurer-Stroh[19,56,57,58], Theresia Handayani Mina[50], Shi Qi Mok[59], Hong Kiat Ng[50], Chee Jian Pua[37], Elio Riboli[43], Tyler Hyungtaek Rim[18,14], Charumathi Sabanayagam[18,14], Wey Cheng Sim[11], Tavintharan Subramaniam[60], Ee Shien Tan[30], Eng King Tan[61,62], Erwin Tantoso[13,19], Darwin Tay[50], Yik Ying Teo[6], Yih Chung Tham[18,14], Li-xian Grace Toh[11], Pi Kuang Tsai[11], Rob M. van Dam[6,63,64], Lavanya Veeravalli[13], Gervais Wansaicheong Khin-lin[65], Andreas Wilm[13], Chengxi Yang[37], Fabian Yap[31,66] & Yik Weng Yew[3,67]

[33]National Supercomputing Centre, Singapore 138632, Singapore. [34]Centre for Genomics and Personalised Health, Genomics Research Centre, QUT Kelvin Grove, Australia. [35]Clinical Research & Innovation Office, Tan Tock Seng Hospital, Singapore 308433, Singapore. [36]Cardiovascular Academic Clinical Program, Duke-NUS Medical School, Singapore 169857, Singapore. [37]National Heart Research Institute Singapore, National Heart Centre Singapore, Singapore 169609, Singapore. [38]National Heart and Lung Institute, Imperial College London, London, UK. [39]Endocrinology, Tan Tock Seng Hospital, Singapore 308433, Singapore. [40]Metabolic Medicine, Lee Kong Chian School of Medicine, Nanyang Technological University, Singapore 308232, Singapore. [41]Health Services and Systems Research, Duke-NUS Medical School, Singapore 169857, Singapore. [42]Cardiovascular Research Institute, National University Health System, Singapore 119228, Singapore. [43]School of Public Health, Imperial College London, London W2 1PG, UK. [44]Department of General Practice and Primary Health Care, University of Helsinki and Helsinki University Hospital, Helsinki 00014, Finland. [45]Folkhälsan Research Center, Folkhälsan 00250, Finland. [46]Cardiovascular Diseases Translational Research Programme, Yong Loo Lin School of Medicine, National University of Singapore, Singapore 119228, Singapore. [47]Department of Endocrinology, Singapore General Hospital, Singapore 168752, Singapore. [48]Yong Loo Lin School of Medicine, National University of Singapore, Singapore 119228, Singapore. [49]Center for Genome Diagnostics, Genome Institute of Singapore, Agency for Science, Technology and Research, Singapore 138672, Singapore. [50]Population and Global Health, Lee Kong Chian School of Medicine, Nanyang Technological University, Singapore 308232, Singapore. [51]Department of Psychosis, Institute of Mental Health, Singapore 539747, Singapore. [52]Department of Haematology-Oncology, National University Cancer Institute Singapore, Singapore 119228, Singapore. [53]Department of Paediatrics, Yong Loo Lin School of Medicine, National University of Singapore, Singapore 119228, Singapore. [54]National Healthcare Group Eye Institute, Tan Tock Seng Hospital, Singapore 308433, Singapore. [55]National Skin Centre, Singapore 308205, Singapore. [56]A*STAR Infectious Diseases Labs, Agency for Science, Technology and Research, Singapore 138648, Singapore. [57]Department of Biological Sciences, National University of Singapore, Singapore 117558, Singapore. [58]National Public Health Laboratory, National Centre for Infectious Diseases, Singapore 308442, Singapore. [59]Laboratory of Complex Disease Genetics, Genome Institute of Singapore, Agency for Science, Technology and Research, Singapore 138672, Singapore. [60]Diabetes Centre, Admiralty Medical Centre, Singapore 730676, Singapore. [61]Neurology Department, National Neuroscience Institute, Singapore 308433, Singapore. [62]Neuroscience and Behavioural Disorders Programme, Duke-NUS Medical School, Singapore 169857, Singapore. [63]Department of Nutrition, Harvard T.H. Chan School of Public Health, Harvard University, Boston, MA 02115, USA. [64]Department of Exercise and Nutrition Sciences, Milken Institute School of Public Health, The George Washington University, Washington, DC 20052, USA. [65]Diagnostic Radiology, Tan Tock Seng Hospital, Singapore 308433, Singapore. [66]Division of Medicine, KK Women's and Children's Hospital, Singapore 229899, Singapore. [67]National Skin Centre, Singapore 308232, Singapore.

