## [Peer Review File · Nature Communications]

Analysis of human disease variants from ancestrally diverse Asian genomesREVIEWER COMMENTS

Reviewer #1 (Remarks to the Author):

Chan et. al. reports on disease carrying variants in 9051 individuals in Singapore, which represents Indian, Chinese, and Malay ancestries. They determine the frequency of clinically significant AD and AR variants and the differences in clinical variation across the three ancestry groups. This manuscript puts in context an important problem in carrier screening that has implications for precision medicine efforts. However, some comments need to be addressed to clarifying the points made in the manuscript.

Major

1. Methods - How were individuals selected to be in each of the six participating studies? And then how were they selected for sequencing? Are all the cohorts truly random samples from the Singapore population?
2. Methods - WGS was done to a target depth of 30X and 15X. The authors evaluated for potential batch effects for variant carrier frequencies. However, the strong correlation in the carrier frequencies seems to be drive by one common variant. One would expect that the difference in target depth to be more relevant for rare variants. What are the correlations after removing common variants? Could in-house gene panels be driving up some of the prevalence estimates?
3. Methods – it seems that local ancestry estimates may help explain some of the differences in self-reported R/E with genetic ancestry given the admixture of some of the individuals and would be more accurate than the definition of ancestry-specific variants used.
4. Results – ST2 states that 82% of individuals carry a PLP variant. This calls in question the definition used for a PLP variants.
5. Results - Is there any difference in the prevalence of AD and AR variants by the 3 sources of genes?
6. Figure 3 is difficult to interpret and may be better displayed as a table.

Minor

7. Figure 1C – Please add in how the normalized Z score was calculated? Normalized Z score of what comparison? Difference in carrier frequencies of P/LP variants across which ancestry groups?
8. Methods – Was any QC applied before creating the dataset for kinship and admixture inference?
9. Methods/Results – for the LDL against LDLR variant status, the methods state that age was adjusted for but the results do not include an age adjustment. Why not use linear regression to predict LDL cholesterol levels from LDLR variant status instead of predicting LDLR variant status from LDL?

10. Results - Please explicitly define PharmGKB level 1 evidence in the manuscript

Reviewer #2 (Remarks to the Author):

This manuscript reports the analysis of 9051 genomes of individuals from Singapore. The authors searched for variants relevant to monogenic disorders as well as pharmacogenomics. Results for monogenic conditions were compared to carrier screening protocols in order to identify variants that should be included to meet screening guidelines. The findings include actionable clinical variants in 3.7%. Over 99% had at least 1 pharmacogenetic variant, and 22.5% of those with CDC Tier 1 genetic conditions had a pharmacogenetic variant in a gene relevant to a medication typically used to treat that disorder. Interestingly, the authors also find a high rate of ancestry-specific (founder) variants that were discordant from individuals' self-reported ancestry.

I was asked to specifically review the pharmacogenetic analyses in the manuscript. Thus, the majority of my review relates to those methods, findings, and discussion. The authors evaluated pharmacogenetic variation across 3 ancestral groups (Chinese, Malay, and Indian [South Asian]). They focused on genes with PharmGKB level 1 evidence, including HLA and CYP2D6 genes. Pharmacogenetic findings include a high frequency of individuals with at least one variant (>99%), with a mean of 5/individual. Much of this was driven by the very high frequency of a single variant in VKORC1, known to be common in Asian individuals, and variants in HLA, which can be life-threatening. Also identified were common variants in genes such as CYP2C19 and CYP2D6. It is a notable strength of the paper that the authors disaggregate the data based on the 3 ancestral groups, as frequencies can be quite different across these groups which are often lumped together as "Asian".

I have several specific comments that I hope can strengthen the manuscript.

Regarding the non-pharmacogenetic analysis and findings:

-Disease variants are denoted with variable/inconsistent nomenclature, and should be standardized as much as possible, while including the commonly used notation for the specific field at first mention.

-Much is made of the variants that were over-represented in these groups, and a few mentions are made in the discussion of which variants were under-represented. A formal analysis of variants that are under-represented may be of interest; For example for the screening tests, are there variants that are so rare in this population, they are more likely to be a false positive than a true positive if identified?

-Do the disease variant frequencies match the expected relative rates of disease based on epidemiologic data?

-T1: Suggest indicating which conditions are AR vs. X linked with a footnote.

Regarding the pharmacogenetic analysis and findings:

-The definition of actionable variants should be made more clear. The use of PharmGKB level 1 designation for the genes included is clear; for the variants within these genes, it appears that they have included variants that have defined functional status leading to an actionable phenotype. This should be made more clear in the methods to enable others to replicate the approach for other populations.

-In the abstract, the report of pharmacogenetic variants in 22.5% that would help guide therapy is so incompletely explained that readers will not know what to do with this information. Please include more detail.

-P14 line 57 "CYP2C19 and CYP2D6" is ambiguous. Do you mean "or"?

-p14 line 47-49 imply that CYP2D6 and CYP2C19 both metabolize the listed drugs, which is not true. Better to state the findings for these 2 genes separately.

-P14 line 59 implies that adverse responses were identified. In actuality, the authors searched for genetically-predicted phenotypes (e.g. CYP2D6 poor metabolizers) that confer increased risk for adverse drug responses. The explanation of these analyses is actually much more clear in the figure legend than in the text.

-P15 lines 71-78: Are the reported pharmacogenetic allele frequencies among those with risk conditions different than those without those conditions?

-For the analysis of actionable phenotypes among those with Tier 1 CDC conditions, how were the target medications identified?

-In general, are the pharmacogenetic variant/risk allele frequencies different than what is reported in available datasets, such as those available from CPIC?

-Were novel variants identified in any of the pharmacogenetic genes?

-A similar analysis could be performed for pharmacogenetic testing as was done for the carrier screening, looking for variants that should be included. Are there pharmacogenetic variants that should be included on pharmacogenetic tests, but are omitted from typical testing platforms and/or from CAP-CLIA lists of recommended variants?

-In the discussion, consider including a discussion of the possibility of including pharmacogenetic test results on disease-centered genetic test result reports. The data presented on the frequency of actionable phenotypes among the Tier 1 conditions argues for this approach.

-T2: Suggest including n with each % for the reported alleles

-T2: Please specify range of activity scores for CYP2D6 (e.g. IM)

-F3: The use of circles, while perhaps aesthetically pleasing, makes it more difficult to compare across the 3 ancestry groups (which could build a case for disaggregation) and in general makes it difficult to actually quantify the data. Suggest having a series of stacked bars, perhaps one of N and one of frequency, to enable better visualization of the data. I do appreciate the inclusion of the n for each group in the center, and stacked bars would also enable enumeration of each ancestral group.

-F3: For simvastatin, consider renaming "statin drugs", as several are included in the current guideline.

Manuscript ID NCOMMS-22-14716A : “Analysis of human disease variants from ancestrally diverse Asian genomes”

We thank the editor and reviewers for taking time out from their busy schedules to provide constructive critique that has strengthened our manuscript. We have highlighted the changes in the ‘revised manuscript with changes’ document based on the reviewers’ comments and have provided responses to each comment below.

IN RESPONSE TO THE REVIEWERS’ COMMENTS:

REVIEWER 1:

COMMENT TO THE AUTHOR:

1. Methods – How were individuals selected to be in each of the six participating studies? And then how were they selected for sequencing? Are all the cohorts truly random samples from the Singapore population?

RESPONSE:

All six studies included in the SG10K_Health cohort were cross-sections of the Singaporean multi-ethnic population, prospectively enrolled for population studies based on respective recruitment protocols referenced in Supplementary Table 1 and are broadly reflective of the Singapore multi-ethnic composition. Blood-derived germline DNA from all recruited participants were included for sequencing, with only samples that failed DNA sequencing quality metrics being excluded from the final dataset.

Whereas carrier frequencies of recessive conditions are not impacted by any potential cohort bias, we recognize that there may be slight biases in the representation of more severe/penetrant autosomal dominant conditions (e.g., individuals with cancer are less likely to participate in the studies), due to the open study recruitment. Nevertheless, our observed prevalence of pathogenic variants in the American College of Medical Genetics and Genomics (ACMG) recommended 73 genes for reporting secondary findings (SF v3.0) is consistent with other cohorts, such as the study in 149,960 UK Biobank participants¹, suggesting that our cohort biases are not substantially different from other studies.

COMMENT TO THE AUTHOR:

2. Methods – WGS was done to a target depth of 30X and 15X. The authors evaluated for potential batch effects for variant carrier frequencies. However, the strong correlation in the carrier frequencies seems to be driven by one common variant. One would expect that the difference in target depth to be more relevant for rare variants. What are the correlations after removing common variants? Could in-house gene panels be driving up some of the prevalence estimates?

RESPONSE:

We have revised our evaluation to remove common variants with carrier frequencies of 10% or more, assessing variants with carrier frequencies less than 10% for both dominant and recessive conditions. As shown in the revised Supplementary Figure 2 below (now

Supplementary Figure 5, panels A and B), samples with 30X and 15X coverage were strongly correlated (Pearson's $r > 0.85$) across all ancestry groups for variants of both dominant and recessive conditions.

Supplementary Figure 5: Strong concordance in gene-level carrier frequencies of samples sequenced to 15X and 30X target depth. Carrier frequencies for (A) autosomal dominant disorder genes and (B) recessive disorder genes were evaluated for 30X versus 15X samples of Chinese (CH, left panel), Indian (IND, middle panel) and Malay (MY, right panel) ancestry. Genes with carrier frequencies $< 10\%$ were included for evaluation. (C) Carrier frequency of pharmacogenomic variants detected using ALDY (left panel), CYRIUS (middle panel) and VCF-derived (right panel) methods were strongly correlated for 30X versus 15X samples across all ancestry groups.

Regardless of sample target depth, all samples were evaluated on the same gene set, which was curated from multiple sources for a comprehensive coverage of genes associated with monogenic disorders. As shown in the figure below, strong correlation (Pearson's $r > 0.93$) in carrier frequencies was observed between 30X and 15X samples even when we limited our evaluation to 73 genes in the ACMG secondary findings (v3.0) genes list, indicating no significant batch effects from samples with different target depth.

COMMENT TO THE AUTHOR:

3. Methods – It seems that local ancestry estimates may help explain some of the differences in self-reported R/E with genetic ancestry given the admixture of some of the individuals and would be more accurate than the definition of ancestry-specific variants used.

RESPONSE:

We appreciate the Reviewer's suggestions and have adopted local ancestry inference to define ancestry-specific variants. In our revised manuscript, we have thus revised our method as recommended:

(Page 27, Line 7-16):

"To estimate local ancestry, we used phased genotypes generated using EAGLE (v2.4.1) and retained only SNPs with minor allele frequency $\geq 1\%$ and call rate of ≥ 0.5 . We selected 100 individuals from each ancestry group with the highest respective ancestral component, and the combined 300 individuals representing Chinese, Indian and Malay ancestry groups were used as the reference panel for inference of local ancestry using RFMix (v2.03-r0)⁷⁷ on default settings. In the analysis of discordant variant carriers in Figure 2B, we defined ancestry-specific variants by the following criteria: 1) P/LP variants with allele count ≥ 5 , and 2) the variant exclusively occurs in an allele with the same inferred local ancestry. For instance, a

Chinese-specific variant is one that occurs exclusively in alleles with inferred local ancestry of Chinese origin.”

We would like to clarify that individual genetic ancestries were inferred from the highest ancestral component estimated by ADMIXTURE for each individual, using K=3 hypothetical ancestral components. We recognize that the lack of clarity in our methods description may have been confusing and have amended, as shown below:

(Page 26, Line 25 – Page 27, Line 5):

“For global ancestry inference, we performed admixture analysis to estimate the proportions of three hypothetical ancestral components in each sample on ADMIXTURE (ver 1.3.0)¹⁵ with K=3 using the same PLINK BED reference panel. The hypothetical components of K=3 has been demonstrated to sufficiently delineate the three major ancestry groups (Chinese, Indian, Malay) in a Singaporean cohort¹⁴. The highest of the three estimated ancestral components for each individual was inferred as genetic ancestry. For the purpose of our analyses, “genetic ancestry” assigned to each individual is a statistical construct calculated from inherited genetic variants and is not equivalent to, nor intended to replace, self-reported race or ethnicity, which are social constructs identified by the individuals.”

Our manuscript reported 268 individuals with inconsistent self-reported race/ethnicity (R/E) with ADMIXTURE-inferred ancestry (i.e. ‘R/E-mismatched group’) and our analysis showed that the discrepant R/E versus genetic ancestry was congruent with the admixed ancestral components estimated by ADMIXTURE, as shown in Supplementary Table 8. For instance, the 8 individuals self-reported as Chinese but were ADMIXTURE-inferred Malay showed an overall appreciable mixture of Chinese and Malay ancestral components.

Supplementary Table 8: Median values of genetic ancestral components of individuals with mismatched self-reported race/ethnicity (R/E) and ADMIXTURE-inferred genetic ancestry.

Self-reported R/E	Admixture-inferred ancestry	No. individuals	Median of genetic ancestral component		
			Chinese	Indian	Malay
Chinese	Malay	8	0.16	0.09	0.65
Indian	Chinese	35	0.50	0.43	0.08
	Malay	27	0.13	0.32	0.52
Malay	Chinese	126	0.55	0.06	0.35
	Indian	44	0.15	0.49	0.31
Others	Chinese	6	0.55	0.08	0.32
	Indian	11	0.10	0.87	0.04
	Malay	11	0.13	0.08	0.82

Using the highest ancestral component proportion (maxQ) as an indicator of admixture, we reported that R/E-mismatched individuals were genetically more admixed, with a median maxQ value of 0.53 compared to R/E-matched individuals (median maxQ = 0.87).

Supplementary Figure 4: Comparison of maxQ values between individuals of R/E-mismatched group and R/E-matched group. Individuals with mismatched self-reported and genetic ancestries are significantly more admixed as indicated by the lower median maxQ, which is a measure of the highest ancestral component proportion. Statistical significance of difference between mismatched and matched groups was evaluated by Wilcoxon rank-sum test.

We discussed that this is likely due to recent admixture, contributed by mixed parentage and driven by cross-ancestry marriages, which are not uncommon in Singapore’s multi-ethnic population (in 2021, 16.2% of marriages in Singapore were inter-ethnic)².

(Page 13, Line 9-14):

“Using the highest ancestral component proportion, maxQ, as a measure of admixture (with lower maxQ indicating higher admixture), we found that the R/E-mismatched group had significantly lower median maxQ compared to R/E-matched group (0.53 vs. 0.87, $P=1.93 \times 10^{-89}$), implying that recent admixture (e.g., mixed parentage), may be prevalent among R/E-mismatched individuals (Supplementary Figure 4).”

(Page 19, Line 20-23):

“This is consistent with Singapore’s history of immigration, epitomized by admixture among the Peranakan community established through inter-marriage between Chinese and Indian immigrants with native Malays since the 15th century⁴⁸.”

COMMENT TO THE AUTHOR:

4. Results – ST2 states that 82% of individuals carry a PLP variant. This calls in question the definition used for a PLP variants.

RESPONSE:

In Supplementary Table 2, the tabulation of 'No. carriers of P/LP variant' reflects the number of individuals harboring at least one variant fulfilling our classification criteria for pathogenic/likely pathogenic (detailed under Methods subsection: Variant classification and interpretation) that occurred in any of the 4,143 genes analyzed. This includes genes associated with autosomal dominant (AD), autosomal recessive (AR) and X-linked monogenic disorders. We expected a high carrier rate of P/LP variants as recessive conditions are not expressed in heterozygous carriers; indeed our finding of 81.6% is consistent with a study of recessive alleles in 6,447 European exomes, which reported that over 85% of individuals harbor at least one recessive P/LP allele, with an average of 2.2 P/LP variants per individual in any AR disorder³.

We have included a footnote in Supplementary Table 2 to clarify the definition of 'P/LP variant' within the context.

COMMENT TO THE AUTHOR:

5. Results – Is there any difference in the prevalence of AD and AR variants by the 3 sources of genes?

RESPONSE:

The curated gene set used in this study was derived from the 3 sources listed in Methods (PanelApp, OMIM, in-house gene panels), with the intention to comprehensively cover the genes associated with human monogenic disorders. It was not our objective to compare prevalence of AD or AR condition variants by the source of genes, but to capture a more comprehensive gene list associated with human disorders for analysis.

COMMENT TO THE AUTHOR:

6. Figure 3 is difficult to interpret and may be better displayed as a table.

RESPONSE:

We thank the Reviewer for the suggestion and have revised Figure 3 for interpretability. We have also included Supplementary Table 13 to tabulate the individual counts for reference.

Figure 3: Pharmacogenetic variants-associated actionable phenotype to drugs used for treatments identified among carriers of germline pathogenic/likely pathogenic (P/LP) variants in a CDC Tier 1 condition. Only pharmacogenetic variant-drug combinations supported by PharmGKB Level 1A/1B evidence were considered. Actionable pharmacogenetic phenotype is defined as CYP2D6 intermediate/poor metabolizers for individuals at-risk of HBOC, UGT1A1 intermediate/poor metabolizers for individuals at-risk of LS, and SLCO1B1 intermediate/poor function for individuals at-risk of FH. HBOC: hereditary breast and ovarian cancer syndrome, LS: Lynch syndrome, FH: familial hypercholesterolemia. The overall proportion of individuals with actionable phenotype within

each ancestry is indicated to the right of each bar chart panel, with number of individuals indicated in parentheses.

COMMENT TO THE AUTHOR:

7. Figure 1C – Please add in how the normalized Z score was calculated? Normalized Z score of what comparison? Difference in carrier frequencies of P/LP variants across which ancestry groups?

RESPONSE:

The colour scale in Figure 1C is the gene-wise normalization of carrier frequency across the three ancestry groups using the following formula, where x is the carrier frequency for each ancestry group, μ is the mean carrier frequency across all three ancestry groups, and σ is the standard deviation:

$$z = (x - \mu) / \sigma$$

We have included the description in the legend of Figure 1C:

(Page 36, Line 8-11):

“(C) Genes of recessive conditions with significant differences in carrier frequency of P/LP variants across ancestry groups. Colour scale maps to row-wise z-scores, obtained by subtracting from each carrier frequency the row average and then dividing the value by the row standard deviation.”

COMMENT TO THE AUTHOR:

8. Methods – Was any QC applied before creating the dataset for kinship and admixture inference?

RESPONSE:

The reference panel generated comprised single nucleotide polymorphisms (SNPs) pruned with $r^2 > 0.5$, using the PLINK recommended settings of window sizes of 50 SNPs with steps of 5 SNPs and variant inflation factor (VIF) of 2. We have updated our methods to reflect the setting used.

(Page 26, Line 17-27):

“To perform kinship analysis, we extracted a set of known polymorphic sites from the full VCF using Somalier (v0.2.13)⁷⁴ and processed using PLINK (v1.90b3.46)⁷⁵ to produce a PLINK BED reference panel, consisting single nucleotide polymorphisms (SNPs) pruned with $r^2 > 0.5$ (using PLINK recommended settings of window sizes of 50 SNPs with steps of 5 SNPs across the genome). We used Kinship-based Inference for Genome-wide association studies (KING, ver 2.2.3)⁷⁶ to calculate pairwise kinship coefficients and considered pairs of samples with kinship coefficient ≥ 0.0884 as related, and randomly select one from each pair for exclusion.

For global ancestry inference, we performed admixture analysis to estimate the proportions of three hypothetical ancestral components in each sample on ADMIXTURE (ver 1.3.0)¹⁵ with $K=3$ using the same PLINK BED reference panel.”

COMMENT TO THE AUTHOR:

9. Methods/Results – for the LDL against LDLR variant status, the methods state that age was adjusted for but the results do not include an age adjustment. Why not use linear regression to predict LDL cholesterol levels from LDLR variant status instead of predicting LDLR variant status from LDL?

RESPONSE:

We noticed the typo highlighted and we apologize for the confusion resulting from incomplete description of our statistical model in Results. We have amended the description for clarity:

(Page 14, Line 19-24):

“We found that individuals harbouring P/LP and VUS-FP variants were more likely to have clinically high LDL cholesterol levels (defined as ≥ 4.1 mmol/L by the Ministry of Health Singapore) compared to non-carriers (Fig. 2D), even after adjusting for age, sex, ancestry and lipid-lowering medication intake (P/LP: OR=10.83, 95%CI=4.52-30.05, $P=5.18 \times 10^{-7}$; VUS-FP: OR=9.67, 95%CI=1.41-190.62, $P=0.044$).”

We also wish to clarify that our logistic regression modelling was aimed at testing whether LDLR variant status could predict for clinically high LDL cholesterol levels (defined as ≥ 4.1 mmol/L by the Ministry of Health Singapore), with correction for covariates including age, sex, ancestry and lipid-lowering medication intake.

We have previously performed linear regression modelling and found that LDLR P/LP and VUS-FP variant carriers had higher cholesterol levels, even after correction for the same covariates (age, sex, ancestry and lipid-lowering medication intake):

LDLR variant status	Coefficient estimate, β	P-value
P/LP	1.08	5.68×10^{-11}
VUS-FP	0.80	0.029

In our manuscript, we opted to report the logistic regression model as we believe that interpretation of odds ratio would be more meaningful from the public health perspective for benchmarking against the clinical guidelines, for the benefit of clinical practitioners among readers.

COMMENT TO THE AUTHOR:

10. Please explicitly define PharmGKB level 1 evidence in the manuscript.

RESPONSE:

We have updated our Methods to define PharmGKB level 1 evidence, as shown below:

(Page 27, Line 19-27):

“For profiling the pharmacogenomic landscape, we consolidated a list of 23 pharmacogenes from the CPIC (Clinical Pharmacogenetics Implementation Consortium) drug-gene pair list (Supplementary Table 16, accessed Aug 30 2021) with Pharmacogenomics Knowledgebase (PharmGKB) clinical annotation level of evidence 1A/1B, which are defined as: (Level 1A) gene-drug combinations with variant-specific prescribing guidance in existing clinical guideline annotation or an FDA-approved drug label annotation, and minimally one publication supporting the clinical annotation, or (Level 1B) gene-drug combinations with no variant-specific prescribing guidance but has a high level of evidence supporting the association with at least two independent publications⁷⁸.”

REVIEWER 2:

COMMENT TO THE AUTHOR:

1. Disease variants are denoted with variable/inconsistent nomenclature, and should be standardized as much as possible, while include the commonly used notation for the specific field at first mention.

RESPONSE:

We thank the Reviewer for the comment and have standardized variant nomenclature to include cDNA and protein change at first mention, followed by referencing the protein change (or cDNA change for splice site/non-coding variants) at subsequent mentions. For pharmacogenetic variants, we included cDNA and protein change for variants with rsID at first mention, followed by rsID at subsequent mentions. Star nomenclatures were retained for pharmacogenes with star nomenclature. For instance:

- *MYBPC3* c.1790G>A (p.Arg597Gln)
- *IL36RN* c.115+6T>C
- *CYP4F2* rs2108622 (c.1297G>A, p.Val433Met)
- *CYP2C9**3

COMMENT TO THE AUTHOR:

2. Much is made of the variants that were over-represented in these groups, and a few mentions are made in the discussion of which variants were under-represented. A formal analysis of variants that are under-represented may be of interest; For example for the screening tests, are there variants that are so rare in this population, they are more likely to be a false positive than a true positive if identified?

RESPONSE:

In clinical practice, genetic testing for monogenic disorders is typically performed using DNA sequencing methods such as single gene or multi-gene panel sequencing; some validated with orthogonal methodologies to reduce reporting of false positives. Screening tests using array-based genotyping of single nucleotide polymorphisms (SNPs) are highly unreliable for rare variants due to the data clustering methodology of SNP genotyping, and thus should not be used to guide clinical decisions.

We agree with the Reviewer that on screening tests using SNP array genotyping, the likelihood of true positive declines with rare variants, which has been demonstrated in a study comparing SNP genotyping and next generation sequencing (NGS) data for 49,908 individuals in the UK Biobank⁴. Using rare (allele frequency < 0.01%) *BRCA* variants, Weedon et. al showed that the positive predictive value of SNP genotyping was only 4.2% and poorly predicted the risk of BRCA-related cancer in UK Biobank participants compared to NGS-based sequencing data (SNP array: OR=1.31, 95% CI=0.99-1.71 versus NGS positive sequencing result: OR=4.05, 95% CI=2.72-6.03). Our study using the NGS-based whole genome sequencing on 9,051 Singaporeans identified 4,960 pathogenic/likely pathogenic (P/LP) variants in monogenic disorders with over 85% (4,236/4,960) carried by only 1-2 individuals, most of which would be incorrectly genotyped if screened using array-based SNP genotyping.

We thank the Reviewer for the comment and have included this discussion point in our revised manuscript:

(Page 20, Line 25-28):

“Beyond diversity, we also showed that monogenic disorder pathogenic variants are mostly rare, with >85% carried in only 1-2 individuals, supporting the need for comprehensive sequence-based testing as opposed to array-based single nucleotide polymorphism (SNP) genotyping⁵².”

COMMENT TO THE AUTHOR:

3. Do the disease variant frequencies match the expected relative rates of disease based on epidemiologic data?

RESPONSE:

The concordance of disease variant frequencies with disease epidemiological data is dependent on whether the disease phenotype is well-characterized and the accuracy of diagnosis, which is particularly challenging for rare disorders with non-specific clinical features. For well-studied conditions such as alpha-thalassemia, the higher carrier frequency of alpha-thalassemia deletional variants (19kb *HBA1/HBA2* SEA deletion, 1.16%) compared to non-deletional variants (point mutations in *HBA1* or *HBA2*, collectively 0.36%) in our study is consistent with the higher incidence of deletional Haemoglobin H (HbH) subtype reported in Singapore (0.6% vs 0.058% non-deletional HbH subtype)⁵. We also observed concordance in the carrier frequency of relative disease incidence across ancestry groups; for instance, the higher incidence of beta-thalassemia among Malays (6.3% vs 2.7% Chinese and 0.7% Indians)⁶ is congruent with the significantly higher carrier frequency of *HBB* pathogenic variants among Malays (7.6% vs 1.7% Chinese and 1.2% Indians) in our dataset. This trend was also true for familial hypercholesterolemia (FH), whereby the majority (>75%) of individuals with FH reported in a Singaporean study are Chinese harbouring pathogenic mutations in *LDLR*⁷, consistent with our finding of significantly higher FH risk among Chinese (1.05% vs 0.15% Indians and 0.25% Malays).

Genetic conditions with variable clinical phenotypes and non-specific symptoms are less accurately detected, resulting in the lack of reliable epidemiological data. For instance, we have shown in this study and in previous reports that the carrier frequencies of the citrin deficiency-linked *SLC25A13* and Wilson disease-linked *ATP7B* pathogenic mutations are unexpectedly high in the Singaporean population (approximately 1-2%)⁸, but the lack of epidemiological data suggests that the non-specific phenotypes of these conditions – ranging from variable liver dysfunction to neuropsychiatric symptoms – may contribute to their underdiagnosis. We believe that the findings from our study are important in narrowing this knowledge gap, so that healthcare practitioners will be cognizant of likely prevalent disorders among Asians that may otherwise go undetected owing to non-specific clinical presentations. For instance, greater awareness of the high carrier frequency of *SLC25A13* in our population owing to our findings have resulted in changes to newborn screening procedures to better identify neonates that may have citrin deficiency.

COMMENT TO THE AUTHOR:

4. T1: Suggest indicating which conditions are AR vs X-linked with a footnote.

RESPONSE:

Of the recessive conditions listed in Table 1, only glucose-6-phosphate dehydrogenase (G6PD) deficiency is X-linked. We have included in the footnote to distinguish the X-linked mode of inheritance of *G6PD* over the remaining autosomal recessive conditions.

COMMENT TO THE AUTHOR:

5. The definition of actionable variants should be made more clear. The use of PharmGKB level 1 designation for the genes included is clear; for the variants within these genes, it appears that they have included variants that have defined functional status leading to an actionable phenotype. This should be made more clear in the methods to enable others to replicate the approach for other populations.

RESPONSE:

We thank the Reviewer for highlighting areas for improvement in our methods, and have amended with details for clarity:

(Page 27, Line 19 – Page 28, Line 20):

“For profiling the pharmacogenomic landscape, we consolidated a list of 23 pharmacogenes from the CPIC (Clinical Pharmacogenetics Implementation Consortium) drug-gene pair list (Supplementary Table 16, accessed Aug 30 2021) with Pharmacogenomics Knowledgebase (PharmGKB) clinical annotation level of evidence 1A/1B, which are defined as: (Level 1A) gene-drug combinations with variant-specific prescribing guidance in existing clinical guideline annotation or an FDA-approved drug label annotation, and minimally one publication supporting the clinical annotation, or (Level 1B) gene-drug combinations with no variant-specific prescribing guidance but has a high level of evidence supporting the association with at least two independent publications⁷⁸. Referencing the CPIC and Pharmacogene Variation Consortium (PharmVar) repositories (accessed April 2021), we identified known pharmacogenetic alleles of these 23 genes using the following methods: (a) Cyrius⁷⁹ (CYP2D6) and Aldy⁸⁰ for genes with star allele nomenclature, (b) HLA-HD⁸¹ for HLA-A and HLA-B alleles, (c) VCF-derived for genes with pharmacogenetic alleles defined by dbSNP rsIDs. Allele frequencies for each allele with a functional status associated with known pharmacogenetic phenotype is tabulated in Supplementary Table 11, whereas the carrier frequency of actionable pharmacogenetic phenotypes associated with the 23 pharmacogenes is tabulated in Table 2, and further consolidated by actionable phenotype with therapeutic recommendation guidelines in Supplementary Table 12. Carrier frequency for diplotypes associated with actionable phenotypes for pharmacogenes with star nomenclature is consolidated in Supplementary Table 17.

For identification of potentially deleterious novel variants (i.e. not found in CPIC or PharmVar), we filtered for LOF variants (frameshift insertions/deletions, nonsense, essential splice site) that: a) are located in MANE transcript, and b) AutoPVS1 indicated PVS1 strength of “Very Strong”, and c) occurred in 10 of the 23 pharmacogenes in our list, for which LOF is the mechanism associated with the actionable phenotype (CYP2B6, CYP2C9, CYP2C19, CYP2D6, DPYD, G6PD, NUDT15, SLCO1B1, TPMT, UGT1A1). Upon manual review, one variant (SLCO1B1 c.1738C>T (p.Arg580), rs71581941) was removed due to poor read coverage.”*

COMMENT TO THE AUTHOR:

6. In the abstract, the report of pharmacogenetic variants in 22.5% that would help guide therapy is so incompletely explained that readers will not know what to do with this information. Please include more detail.

RESPONSE:

We regret the incomplete description due to word limit, and have revised the abstract for clarity, excerpt below:

(Page 5, Line 11-15):

“We profile 23 pharmacogenes with high-confidence gene-drug associations and find 22.4% of Asians at-risk of Centers for Disease Control and Prevention Tier 1 genetic conditions concurrently harbour pharmacogenetic variants with actionable phenotypes, highlighting the benefits of pre-emptive pharmacogenomics.”

COMMENT TO THE AUTHOR:

7. P14 line 57 “CYP2C19 and CYP2D6” is ambiguous. Do you mean “or”?
8. P14 line 47-49 imply that CYP2D6 and CYP2C19 both metabolize the listed drugs, which is not true. Better to state the findings for these 2 genes separately.

RESPONSE:

We thank the Reviewer for the suggestion and have revised the paragraph for clarity:

(Page 15, Line 22 – Page 16, Line 8):

*“Overall, we observed that individuals with actionable pharmacophenotypes associated with commonly prescribed drugs were relatively prevalent, irrespective of ancestry (Table 2). Notably, high fractions of individuals were identified with a genotype affecting the activity of cytochrome P450 family of enzymes (Supplementary Table 12); for instance 51.0%-77.2% individuals across ancestries harboured alleles associated with actionable phenotypes in CYP2C19, which is important for metabolism of widely used drugs including the antiplatelet clopidogrel, antiemetics (proton pump inhibitors) and antidepressants such as selective serotonin uptake inhibitors (SSRIs), whereas 31.1%-47.2% individuals carried actionable pharmacogenetic variants in CYP2D6 for a broad range of drug interactions including opioids, antidepressants, and tamoxifen therapy for cancer. However, we also found that the prevalence of certain pharmacophenotypes was variable by ancestry; for instance, there were significantly more poor metabolizers among Indians (17.4%) compared to Chinese (3.2%, $P=7.28 \times 10^{-66}$) and Malays (1.3%, $P=6.50 \times 10^{-51}$) for UGT1A1, which metabolizes irinotecan-based drugs frequently used in cancer treatments, due to a higher allele frequency of UGT1A1*28 among Indians.”*

COMMENT TO THE AUTHOR:

9. P14 line 59 implies that adverse responses were identified. In actuality, the authors searched for genetically-predicted phenotypes (e.g. CYP2D6 poor metabolizers) that confer increased risk for adverse drug responses. The explanation of these analyses is actually much more clear in the figure legend than in the text.

RESPONSE:

We agree with the Reviewer and have amended the text for clarity:

(Page 16, Line 14-21):

“Next, we explored the intersection of individual genetic disease risk with pharmacogenomic profile by estimating the frequency of individuals harbouring pharmacogenetic variants associated with an actionable phenotype to drugs used for the disorder they are genetically predisposed to. We identified 143 individuals at risk of CDC Tier 1 genetic conditions (HBOC, Lynch syndrome, FH)³⁷, of whom 32 (22.4%) concurrently harboured a pharmacogenetic variant with actionable phenotype to drugs commonly used for treatment of their condition (Fig. 3, Supplementary Table 13).”

COMMENT TO THE AUTHOR:

10. P15 lines 71-78: Are the reported pharmacogenetic allele frequencies among those with risk conditions different that those without those conditions?

RESPONSE:

Using Fisher’s exact test, we observed no significant differences in the frequency of pharmacogenomic genotypes linked to actionable phenotypes between individuals at-risk of CDC Tier 1 genetic conditions compared to those without genetic risk of those conditions, as tabulated below:

CDC T1 genetic condition	Condition status [^]	Actionable PGX phenotype				P-value
		Risk allele carriers		Non-risk allele carriers		
		No.	%	No.	%	
Hereditary breast & ovarian cancer syndrome	Not at-risk (n=6430)	2663	41.4	3767	58.6	0.1745
	At-risk (n=45)	14	31.1	31	68.9	
Lynch syndrome	Not at-risk (n=7018)	3285	46.8	3733	53.2	0.8078
	At-risk (n=16)	8	50.0	8	50.0	
Familial hypercholesterolemia	Not at-risk (n=8774)	1682	19.2	7092	80.8	0.6299
	At-risk (n=63)	10	15.9	53	84.1	

[^]: n in parenthesis indicates number of individuals with genotype available.

CDC T1: Centers for Disease Control and Prevention Tier 1, PGX: pharmacogenomic

COMMENT TO THE AUTHOR:

11. For the analysis of actionable phenotypes among those with Tier 1 CDC conditions, how were the target medications identified?

RESPONSE:

From our list of 23 pharmacogenes with PharmGKB level 1A/1B evidence in gene-drug interaction, we selected the drug most commonly used for treating each CDC Tier1 genetic condition as an illustration.

COMMENT TO THE AUTHOR:

12. In general, are the pharmacogenetic variant/risk allele frequencies different that what is reported in available datasets, such as those available from CPIC?

RESPONSE:

Overall, the allele frequencies observed in our study are consistent with data obtained from CPIC or Allele Frequency Net Database⁹ (for HLA risk alleles) for the closest matched ancestry/ethnic group. For instance, the frequency of risk alleles identified in Chinese and Indians for *VKORC1* rs9923231 and *CYP4F2* rs2108622 were broadly similar to the frequencies reported for East Asian and South Asian groups respectively, in the CPIC guidelines for warfarin-dosing¹⁰:

Risk allele	SG10K_Health		CPIC	
	Chinese	Indian	East Asian	South Asian
VKORC1 : rs9923231	88.6%	15.4%	86.7%	15.6%
CYP4F2 : rs2108622	23.4%	43.2%	23.3%	40.3%

Due to underrepresentation of the Malay ancestry group in most databases, data is lacking for comparison; nevertheless our observed frequencies for selected genes were broadly consistent with previous reports from smaller studies involving the Malay ethnic group^{11,12}.

COMMENT TO THE AUTHOR:

13. Were novel variants identified in any of the pharmacogenetic genes?

RESPONSE:

Using our variant curation workflow, we performed an unbiased search for novel loss-of-function (LOF) variants, now detailed in the revised Methods excerpt below:

“For identification of potentially deleterious novel variants (i.e. not found in CPIC or PharmVar), we filtered for LOF variants (frameshift insertions/deletions, nonsense, essential splice site) that: a) are located in MANE transcript, and b) AutoPVS1 indicated PVS1 strength of “Very Strong”, and c) occurred in 10 of the 23 pharmacogenes in our list, for which LOF is the mechanism associated with the actionable phenotype (CYP2B6, CYP2C9, CYP2C19, CYP2D6, DPYD, G6PD, NUDT15, SLCO1B1, TPMT, UGT1A1). Upon manual review, one

variant (SLCO1B1 c.1738C>T (p.Arg580), rs71581941) was removed due to poor read coverage.”*

We identified a total of 47 putative LOF variants, all with a minor allele frequency (MAF) less than 1%. More than half (33/47, 70.2%) of these variants are rare, occurring in only 1-2 carriers. This is consistent with other pharmacogenomic studies, including whole genome/exome sequencing of 2240 Estonians and 1116 Hong Kong Chinese, reporting more than half (>58%) putative LOF were singletons or doubletons^{13,14}. Our findings together with other genome/exome-scale studies collectively showed that many potentially deleterious pharmacogenetic variants are rare and will go undetected in targeted genotyping assays, highlighting the power of using whole genome sequencing for pharmacogenetic testing.

We have revised our manuscript to include the analysis and discussion for novel pharmacogenetic variants, excerpt below:

(Page 17, Line 2-14):

“To evaluate for potentially deleterious novel pharmacogenetic variants, we curated for LOF variants in 10 of our list of 23 pharmacogenes, whereby LOF is the mechanism associated with actionable phenotype. We identified 47 putative LOF variants, all with a minor allele frequency (MAF) less than 1%. Over half (33/47, 70.2%) of these putative LOF variants are rare, occurring as singletons or doubletons (Supplementary Table 14), consistent with the proportions of singleton-doubleton LOF variants reported in whole genome/exome studies from other populations (> 58%)^{39,40}. Notably, half (25/47, 53.2%) of the putative LOF variants were found within the highly polymorphic CYP2C subfamily of cytochrome P450 genes (CYP2C9, CYP2C19, CYP2D6), in a total of 95 individuals. The large fraction of rare known risk variants and putative LOF variants identified in pharmacogenes important for metabolizing a broad range of drugs suggests that next-generation sequencing-based assays are warranted for comprehensive pharmacogenetic testing, as genotyping assays may miss or inaccurately detect such rare variants.”

COMMENT TO THE AUTHOR:

14. A similar analysis could be performed for pharmacogenetic testing as was done for the carrier screening, looking for variants that should be included. Are there pharmacogenetic variants that should be included on pharmacogenetic tests, but are omitted from typical testing platforms and/or from CAP-CLIA lists of recommended variants?

RESPONSE:

We reviewed the list of known risk alleles linked to actionable phenotypes detected in our cohort (Supplementary Table 11) and found that commercial pharmacogenetic testing gene panels broadly cover all the variants with a minor allele frequency (MAF) > 1%. There were 35 variants with MAF < 0.5% in important pharmacogenes (including CYP2C9, CYP2C19, CYP2D6, DPYD, G6PD, CACNA1S) that were not covered in the commercial testing panels surveyed, supporting the use of next generation sequencing (NGS)-based assays over array-based genotyping for screening.

We revised our manuscript to highlight this point, under Results section:

(Page 15, Line 10-14):

“Of 154 pharmacogenetic variants with actionable phenotype identified (Supplementary Table 11), 76.6% (118/154) had a minor allele frequency (MAF) < 1% and 31.8% (49/154) were very rare variants with carried by only 1-2 individuals, over half (57.1%, 28/49) of which were found in genes of the cytochrome P450 CYP2 family.”

(Page 17, Line 4-14):

“We identified 47 putative LOF variants, all with a minor allele frequency (MAF) less than 1%. Over half (33/47, 70.2%) of these putative LOF variants are rare, occurring as singletons or doubletons (Supplementary Table 14), consistent with the proportions of singleton-doubleton LOF variants reported in whole genome/exome studies from other populations (> 58%)^{39,40}. Notably, half (25/47, 53.2%) of the putative LOF variants were found within the highly polymorphic CYP2C subfamily of cytochrome P450 genes (CYP2C9, CYP2C19, CYP2D6), in a total of 95 individuals. The large fraction of rare known risk variants and putative LOF variants identified in pharmacogenes important for metabolizing a broad range of drugs suggests that next-generation sequencing-based assays are warranted for comprehensive pharmacogenetic testing, as genotyping assays may miss or inaccurately detect such rare variants.”

COMMENT TO THE AUTHOR:

15. In the discussion, consider including a discussion of the possibility of including pharmacogenetic test results on disease-centered genetic test result reports. The data presented on the frequency of actionable phenotypes among the Tier 1 conditions argues for this approach.

RESPONSE:

We thank the Reviewer for the suggestion and have included the following in our discussion:

(Page 20, Line 15-18):

“This highlights that a substantial fraction of genetically susceptible individuals could benefit from pre-emptive pharmacogenomics to optimize their therapeutic treatments and avoid severe toxicities, indicating opportunities to forge a more comprehensive clinical care by combining pharmacogenomics and genetic disease testing.”

COMMENT TO THE AUTHOR:

16. T2: Suggest including n with each % for the reported alleles.

RESPONSE:

We have updated Table 2 with accompanying number of risk allele carriers identified and the total number of individuals with verified genotype for each gene.

COMMENT TO THE AUTHOR:

17. T2: Please specify the range of activity scores for CYP2D6 (e.g. IM)

RESPONSE:

Table 2 is now amended with activity score range for CYP2D6 referenced from CPIC¹⁵.

Gene	Allele(s)	Genotype	Phenotype
CYP2D6	increased function: *1x2, *1x3, *2x2	activity score >2.0	Ultrarapid metabolizer
	decreased function: *9, *10, *10x2, *10x2+*83, *14, *17, *29, *41, *49, *49x2, *36-*10, *36x2-*10, *36x2-*10-*83, *36+*10x2+*83	activity score ≥0.5 to ≤ 1.0	Intermediate metabolizer
	no function: *3, *4, *4N, *4+*4N, *4+*68, *5, *6, *7, *13, *15, *21, *21x2, *31, *36, *36x2, *36x3, *68, *69, *99, *101, *114	activity score 0 to < 0.5	Poor metabolizer

COMMENT TO THE AUTHOR:

18. F3: The use of circles, while perhaps aesthetically pleasing, makes it more difficult to compare across the 3 ancestry groups (which could build a case for disaggregation) and in general makes it difficult to actually quantify the data. Suggest having a series of stacked bars, perhaps one of N and one of frequency, to enable better visualization of the data. I do appreciate the inclusion of the n for each group in the center, and stacked bars would also enable enumeration of each ancestral group.

RESPONSE:

We thank the Reviewer for the suggestion and have revised Figure 3 for interpretability. We have also included Supplementary Table 13 to tabulate the individual counts for reference.

Figure 3: Pharmacogenetic variants-associated actionable phenotype to drugs used for treatments identified among carriers of germline pathogenic/likely pathogenic (P/LP) variants in a CDC Tier 1 condition. Only pharmacogenetic variant-drug combinations supported by PharmGKB Level 1A/1B evidence were considered. Actionable pharmacogenetic phenotype is defined as CYP2D6 intermediate/poor metabolizers for individuals at-risk of HBOC, UGT1A1 intermediate/poor metabolizers for individuals at-risk of LS, and SLCO1B1 intermediate/poor function for individuals at-risk of FH. HBOC: hereditary breast and ovarian cancer syndrome, LS: Lynch syndrome, FH: familial hypercholesterolemia. The overall proportion of individuals with actionable phenotype within each ancestry is indicated to the right of each bar chart panel, with number of individuals indicated in parentheses.

COMMENT TO THE AUTHOR:

19. F3: For simvastatin, consider renaming “statin drugs”, as several are included in the current guideline.

RESPONSE:

We have amended the mention of simvastatin to “statin drugs” in Figure 3, as recommended.

REFERENCES

1. Halldorsson, B. V. *et al.* The sequences of 150,119 genomes in the UK Biobank. *Nature* (2022) doi:10.1038/s41586-022-04965-x.
2. Department of Statistics Singapore. *Statistics on Marriages and Divorces, 2021*. www.singstat.gov.sg.
3. Fridman, H. *et al.* The landscape of autosomal-recessive pathogenic variants in European populations reveals phenotype-specific effects. *Am. J. Hum. Genet.* **108**, 608–619 (2021).
4. Mn, W. *et al.* Use of SNP chips to detect rare pathogenic variants: retrospective, population based diagnostic evaluation. *BMJ* **372**, n214 (2021).
5. Ang, S. H. *et al.* Haemoglobin H disease and outcomes in Singapore. *Ann. Acad. Med. Singapore* **51**, 244–246 (2022).
6. Lee, S. Y. *et al.* Evaluation of Thalassaemia Screening Tests in the Antenatal and Non-Antenatal Populations in Singapore. *Ann. Acad. Med. Singapore* **48**, 5–15 (2019).
7. Pek, S. L. T. *et al.* Spectrum of mutations in index patients with familial hypercholesterolemia in Singapore: Single center study. *Atherosclerosis* **269**, 106–116 (2018).
8. Bylstra, Y. *et al.* Population genomics in South East Asia captures unexpectedly high carrier frequency for treatable inherited disorders. *Genet. Med. Off. J. Am. Coll. Med. Genet.* **21**, 207–212 (2019).
9. Gonzalez-Galarza, F. F. *et al.* Allele frequency net database (AFND) 2020 update: gold-standard data classification, open access genotype data and new query tools. *Nucleic Acids Res.* **48**, D783–D788 (2020).

10. Johnson, J. A. *et al.* Clinical Pharmacogenetics Implementation Consortium (CPIC) Guideline for Pharmacogenetics-Guided Warfarin Dosing: 2017 Update. *Clin. Pharmacol. Ther.* **102**, 397–404 (2017).
11. Goh, L. L., Lim, C. W., Sim, W. C., Toh, L. X. & Leong, K. P. Analysis of Genetic Variation in CYP450 Genes for Clinical Implementation. *PLoS One* **12**, e0169233 (2017).
12. Sivadas, A., Salleh, M. Z., Teh, L. K. & Scaria, V. Genetic epidemiology of pharmacogenetic variants in South East Asian Malays using whole-genome sequences. *Pharmacogenomics J.* **17**, 461–470 (2017).
13. Tasa, T. *et al.* Genetic variation in the Estonian population: pharmacogenomics study of adverse drug effects using electronic health records. *Eur. J. Hum. Genet. EJHG* **27**, 442–454 (2019).
14. Yu, M. H. C. *et al.* Actionable pharmacogenetic variants in Hong Kong Chinese exome sequencing data and projected prescription impact in the Hong Kong population. *PLoS Genet.* **17**, e1009323 (2021).
15. Goetz, M. P. *et al.* Clinical Pharmacogenetics Implementation Consortium (CPIC) Guideline for CYP2D6 and Tamoxifen Therapy. *Clin. Pharmacol. Ther.* **103**, 770–777 (2018).

REVIEWERS' COMMENTS

Reviewer #1 (Remarks to the Author):

I appreciate the authors responses to my concerns and the incorporation of local ancestry to define ancestry-specific variation. I have no further concerns.

Reviewer #2 (Remarks to the Author):

The authors have made significant revisions to the manuscript, and they have adequately addressed each of my comments from the primary review.